# Association between carbohydrate quality and glycemic control in individuals with type 1 diabetes: A cross-sectional and meal-based analysis

Lingling Bian[1,2]⊕, Chun Yang🆔[1]⊕, Yanjun Jin[1]⊕, Hong Wang[1], Min Zhu[1], Jingjing Xu[1], Jianling Bai[3], Mei Zhang[1], Yanmei Liu[2], Tao Yang🆔[1]*, Yong Gu[1]*, Hechun Liu🆔[1]*

**1** Department of Endocrinology, The First Affiliated Hospital with Nanjing Medical University, Nanjing, China, **2** Department of Endocrinology, The First people's Hospital of Yancheng, Yancheng, China, **3** Department of Biostatistics, School of Public Heath, Nanjing Medical University, Nanjing, China

⊕ These authors contributed equally to the work
* yangt@njmu.edu.cn; yong.gu@njmu.edu.cn; shipinliuhechun@163.com

## Abstract

The aim of the study was to investigate the association between the quality of carbohydrates in the diet and glycemic control in individuals with Type 1 Diabetes (T1D). In this cross-sectional study, a dietary survey involving 155 individuals with T1D used a food frequency questionnaire (FFQ) and 3-day dietary records from 65 individuals to analyze 538 meals. The relationship between short-term dietary factors and postprandial glycemic fluctuations was evaluated by a mixed-effects model. The FFQ survey indicated that a higher intake of dietary fiber was associated with better long-term glycemic control (HbA1c≤6.5%) in T1D individuals (OR=1.101, p = 0.031). Analysis of 3-day dietary records and continuous glucose monitoring (CGM) data showed an inverse correlation between whole grain intake and postprandial glycemic variability, measured by standard deviation (SD), large amplitude of glycemic excursions (LAGE) and mean amplitude of glycemic excursions (MAGE) (Est. = −0.83, p < 0.01; Est. = −2.4, p < 0.01; Est. = −1.47, p = 0.04). Specifically, blood glucose fluctuations (SD, LAGE, MAGE) were more significant after lunch (p < 0.05), and these fluctuations were negatively related to the intake of whole grains (Est. = −0.45, p = 0.02; Est. = −1.52, p = 0.01; Est. = −1.39, p < 0.01). From a long-term glycemic control perspective, higher dietary fiber intake appears to be associated with improved HbA1c levels, while in terms of short-term glycemic variability, increased whole grain consumption is associated with reduced glucose fluctuations. However, as a cross-sectional analysis, these findings represent observational associations rather than causal evidence. Further validation through prospective cohorts and randomized trials is needed to assess clinical applicability.

**Data availability statement:** All relevant data are within the paper and its Supporting Information files.

**Funding:** This research was funded by the Noncommunicable Chronic Diseases-National Science and Technology Major Project (NO. 2023ZD0507400, 2023ZD0507401, 2023ZD0507403), National Natural Science Foundation of Jiangsu Province (NO. BK20220708) and National Natural Science Foundation of China (NO. 82170837, 82230028, 82404271). The funders had no role in study design, data collection and analysis, decision to publish, or preparation of the manuscript.

**Competing interests:** The authors have declared that no competing interests exist.

## Introduction

Type 1 diabetes (T1D) is an autoimmune condition characterized by the destruction of pancreatic β cells and absolute insulin deficiency, necessitating lifelong insulin replacement therapy for individuals with T1D [1–3].The importance of glycemic control in preventing diabetic complications was initially demonstrated by the Diabetes Control and Complications Trial (DCCT), leading to a recommendation for individuals with T1D to achieve target HbA1c levels of <7.0% to minimize complication risk [4–7]. However, achieving optimal glycemic control continues to pose a significant challenge for individuals with T1D, even in the era of artificial pancreas technology, particularly during the postprandial period [8]. While HbA1c is a crucial indicator of glycemic exposure over the previous 8–12 weeks, it does not capture day-to-day or within-day fluctuations in glucose levels. Moreover, individuals with similar HbA1c values may still experience significantly different rates of complications. For example, in the DCCT cohort, only 11% of the variation in retinopathy risk can be attributed to total glycemic exposure (HbA1c and duration of diabetes) [9,10]. Limitations regarding HbA1c highlight the need for supplementary approaches to evaluate glucose levels in individuals with diabetes. Continuous blood glucose monitoring (CGM) can compensate for this limitation. Time in range (TIR) and other derived indicators from CGM, such as Standard Deviation (SD), Coefficient of Variation (CV), Largest Amplitude of Glycemic Excursions (LAGE), Mean Amplitude of Glycemic Excursions (MAGE), better reflect short-term glycaemic variation and effectively address the limitations of HbA1c [11,12].

Dietary carbohydrate, being the primary macronutrient metabolized into glucose, is the key factor affecting postprandial blood glucose and insulin needs [13]. The long-term low-carbohydrate diet pattern is currently controversial [14]. While they may aid short-term glycemic control, some evidence suggests that sustained, very low-carbohydrate intake could be associated with increased mortality risk. [15,16]. The 2024 ADA guidelines recommend prioritizing high-quality, nutrient-rich sources of carbohydrate that are high in fiber and minimally processed, regardless of the amount of carbohydrate in a meal plan [17]. Dietary fiber, glycemic index (GI), glycemic load (GL) and whole grain are indicators of the quality of dietary carbohydrates [18–21]. There is evidence suggesting that low GI and low GL diets can assist in reducing the risk of T2D by minimizing blood glucose fluctuations [22–25]. Additionally, substituting refined grains with whole grains has been demonstrated to be beneficial for those with T2D [26]. However, the effectiveness of these diets in enhancing glycemic control for T1D remains uncertain and demands further research.

Above all, the current study focuses on the quality of dietary carbohydrates, specifically on dietary fiber intake, as well as whole grain consumption, in order to investigate the relationship between the quality of carbohydrates in the diet and glycemic control in individuals with T1D.

## Materials and methods

### Participants

Individuals with T1D were enrolled from the specialized cohort for T1D in Jiangsu Province Hospital between 30/09/2022 and 31/08/2024.

These participants were diagnosed with T1D by physicians based on the 2020 ADA diagnostic criteria [27]. Inclusion criteria were as follows: 1) Individuals who consent to participate in the study and provide informed consent; 2) Age range: 10–65 years; 3) Confirmed diagnosis of T1D according to ADA2020 standards; 4)Requiring exogenous insulin therapy; 5) HbA1c level less than 12%. Exclusion criteria included pregnancy, celiac disease, and any other illness besides diabetes that significantly affected their health status, for instance, individuals with clinically unstable hyper- or hypothyroidism were excluded to minimize confounding, e.g., thyroid-stimulating hormone [TSH] levels <0.2 µIU/mL or >4.2 µIU/mL. In addition to the primary inclusion/exclusion criteria, participants were required to wear CGM devices for 7 consecutive days and complete three weighed-food dietary records (2 weekdays + 1 weekend day) during the middle 5 days, including submission of food photographs to researchers.

## Ethical consideration

This study was conducted in compliance with the ethical standards outlined in the Declaration of Helsinki and received approval from the Ethics Commission of Jiangsu Province Hospital(2022-SR-481). All subjects provided written informed consent prior to participating in this study, with parental consent obtained for underage participants.

## Data collection

The ARIANT II Hemoglobin Test System (Bio-Rad) was used to measure hemoglobin A1c (HbA1c). The International Physical Activity Questionnaire (IPAQ) [28] was used to evaluate physical activity.

## Dietary assessment

The nutrition survey was conducted in two steps. First, all 155 subjects completed the validated food-frequency questionnaire (FFQ) at the baseline clinical visit. Subsequently, a total of 65 participants maintained a 3-day food record while wearing CGM to further explore the correlation between dietary intake and postprandial glycemic variability at the home stage. Participants were provided with detailed instructions and a completed example page for reference. They were instructed to meticulously document all food and beverage consumption during the recording period, emphasizing the continuation of their usual dietary habits. During the study, participants were instructed to document the time of the first bite of food, composition, and portion size of their meals, as well as promptly capture photographic evidence for researchers' review. The Chinese Food Composition Tables [29] were used to estimate nutrient content and determine GI values for most food items. The classification of grain foods was primarily based on the "Food Composition in China (2009)" [29]; A food item was classified as a "whole grain" if whole grains were listed as the first ingredient on its packaged food label (as per the database) or, for non-packaged items, if it was explicitly defined as such in the database (e.g., brown rice, oatmeal, whole-wheat bread). Conversely, items were classified as "refined grain" if the primary grain ingredient was not a whole grain (e.g., white bread, white rice, pasta made from refined flour). For mixed-grain products (e.g., breads containing a blend of whole-wheat and white flour), we relied on the database's classification and the ingredient list hierarchy. If whole grains were not the first ingredient, the item was classified as refined grain. The quantity of whole grains consumed was estimated based on the portion size reported by the participants and the proportional content provided by the food composition database.

## CGMS

The glycemic variability of T1D individuals was assessed using a Continuous Glucose Monitor (Anytime CT15) for a standardized 7-day monitoring period. During the CGM period, participants completed three 24-hour dietary records, including 2 weekdays and 1 weekend day to capture habitual intake variations during the middle 5-day monitoring period to assess typical intake patterns, including precise portion measurements. At the end of the experimental period, data from CGM were downloaded by dedicated informatics platforms.

## Statistical analysis

R Studio, version 2022.07.0 was used for all statistical analyses. Two-tailed P values <0.05 were considered statistically significant. Categorical variables are presented as frequencies (%), parametric data are presented as mean (SD), and non-parametric data are presented as median [interquartile range].

Spearman correlation coefficient was calculated to examine unadjusted correlations between variables. Multivariate linear regression and logistic regression analyses were conducted to identify dietary factors independently correlated with HbA1c, with HbA1c as the dependent variable and carbohydrate (%E), fat (%E), protein (%E), fiber(g/d) as independent variables. The analyses adjusted for age, duration, BMI, physical activity, insulin dose (IU/kg), and intake of other macro-nutrients to fit different models.

Differences between the type of meal (breakfast, lunch and dinner) in dietary composition, GI, GL, Whole-grain (%Carbs), Refined-grain (%Carbs) and CGM metrics were analyzed using mixed-effect model testing for time interaction. Significant differences were explored with post hoc Bonferroni test for pairwise comparisons. The associations between dietary composition and CGM metrics were assessed by mixed-effect models. To fit mixed-effect models, the lmer function in the lme4 R package was used. Our models included random intercepts to account for the nested structure of the data and the non-independence of repeated measures. Specifically, we included: A random intercept for Subject ID to capture baseline differences between individuals. A random intercept for Day nested within Subject ID to capture day-to-day variability within the same individual.Given the complexity of the model and our sample size, we did not include any random slopes for the dietary variables. We assumed that the fixed effects of the dietary variables are consistent across all subjects and days.

Multiple Correspondence Analysis (MCA) was employed to examine the relationship and structure among multiple categorical variables, encompassing dietary factors and glycemic control indicators.

Given the large number of statistical tests performed in this exploratory analysis, the findings, particularly those with p-values between 0.01 and 0.05, should be interpreted with caution and require independent confirmation in future studies.

We use Multiple Imputation by Chained Equations (MICE) under the assumption that the data were missing at random (MAR). The imputation was performed using the mice package in R. We generated m = 50 imputed datasets to ensure the efficiency of the estimates. The imputation model included all variables used in the primary analysis as well as auxiliary variables that were predictive of the missingness or the missing values themselves. The algorithm ran for 20 iterations. All statistical models were fitted to each of the 50 imputed datasets, and the results were pooled into a single set of estimates using Rubin's rules.

## Results

### Baseline features

From July 2022 to August 2024, a total of 155 T1D individuals from the specialized cohort for T1D at Jiangsu Province Hospital completed FFQ (Fig 1). The baseline characteristics of the participants were summarized in S1 Table in S1 File. The intake of dietary fiber fell well below guideline recommendations [19]. In addition, refined grains were the predominant source of staple foods, which contradicted the recommendation in the Chinese Diabetes Medical Nutrition Treatment guidelines (2022) [30] (Table 1).

### Dietary fiber and glycemic control: Higher dietary fiber intake was associated independently with lower HbA1c levels in individuals with T1D

The relationship between HbA1c and meal composition is presented in Fig 2. There was a significant inverse association between HbA1c and fiber intake (Fig 2a, Fig 2b, r = −0.27, p < 0.001). The association between dietary fiber and HbA1c

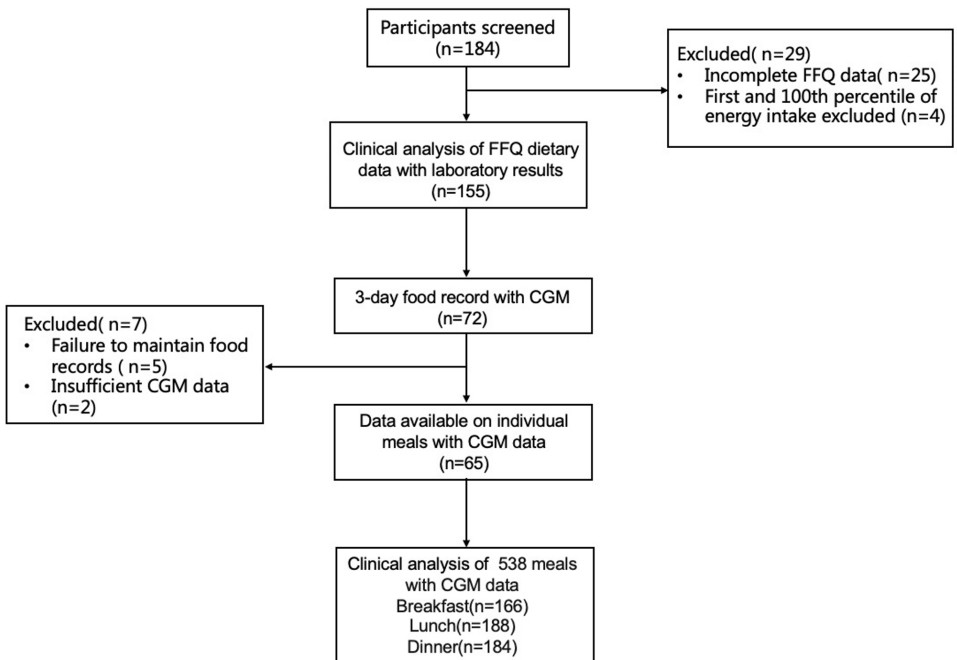

**Fig 1. Study design and analysis diagram.**

remained after controlling for other dietary factors, as well as age, disease duration, physical activity, and insulin dose (Est. = −0.060, p = 0.002, Table 2). Binary logistic regression analysis revealed that after adjusting for other dietary factors as well as age, gender, duration of diabetes, physical activity, and insulin dose, dietary fiber intake emerged as a protective factor for glycemic control (HbA1c≤6.5%) (OR=1.101, p = 0.031, Table 3, Fig 2c).

### Short-term diet-glycemia relationship: The correlation between carbohydrate quality and CGM parameters based on 3-day dietary records

A 3-day dietary record matched with CGM was conducted to investigate the relationship between dietary factors and glycemic fluctuations (Fig 1). S2 Table in S1 File presented the dietary records of 65 participants, encompassing a total of 188 days of nutrient intake and indicators for glycemic control derived from CGM. The overall energy intake of this cohort of individuals with T1D was relatively low (S1 Fig in S1 File), and the intake of dietary fiber was insufficient (S2 Table in S1 File).

The MCA plot divided the characteristics of the population into four quadrants (Fig 3). In the portion of individuals with better glycemic control (TIR ≥ 70%, MAGE<3.9, LAGE<7, SD < 1.5) (quadrant IV), the composition of their diet possessed the following features: having a moderate carbohydrate, E% (45–50%), a higher intake of dietary fiber (Fiber > 15g/d), and a higher proportion of whole grains in the source of carbohydrates (≥40%). Thus, it can be observed that high-quality carbohydrates in the diet of individuals with T1D were associated with better glycemic control.

In the mixed-effect model, there was a negative correlation between fiber intake and glycemic variability, including SD, LAGE and MAGE (Est. = −0.04, p < 0.01; Est. = −0.13, p < 0.01; Est. = −0.08, p < 0.01; Table 4). After further adjustment for age, duration, BMI, insulin dose and physical activity, the results remained statistically significant. GI and GL demonstrated a notable positive association with SD (Table 4). However, there was no significant correlation between CGM parameters (TIR, SD, LAGE, MAGE) and carbohydrate %E (Table 4; S2 Fig in S2 File). Furthermore, analyses revealed significant inverse relationships between the whole grain percentage and key markers of glycemic variability, including

**Table 1. Distribution of dietary nutrients in individuals with T1D.**

| Meal composition | Mean (SD)/ median [IQR] | MNT of diabetes in Chinese guidelines |
|---|---|---|
| Carbs, %E | 47.30 (8.50) | 45-60 |
| Fat, %E | 35.00 [31.00, 41.00] | 20-35 |
| Protein, %E | 17.00 [15.93, 19.00] | 15-20 |
| Fiber, g/d | 10.32 [6.95, 15.70] | 25-36 |
| Refined-grain, g/d | 137.86 [86.97, 192.86] | / |
| Whole-grain, g/d | 15.71 [3.39, 31.96] | / |

Data are presented as mean (SD) for continuous variables with normal distribution, median [interquartile range] for continuous variables that were skewed.

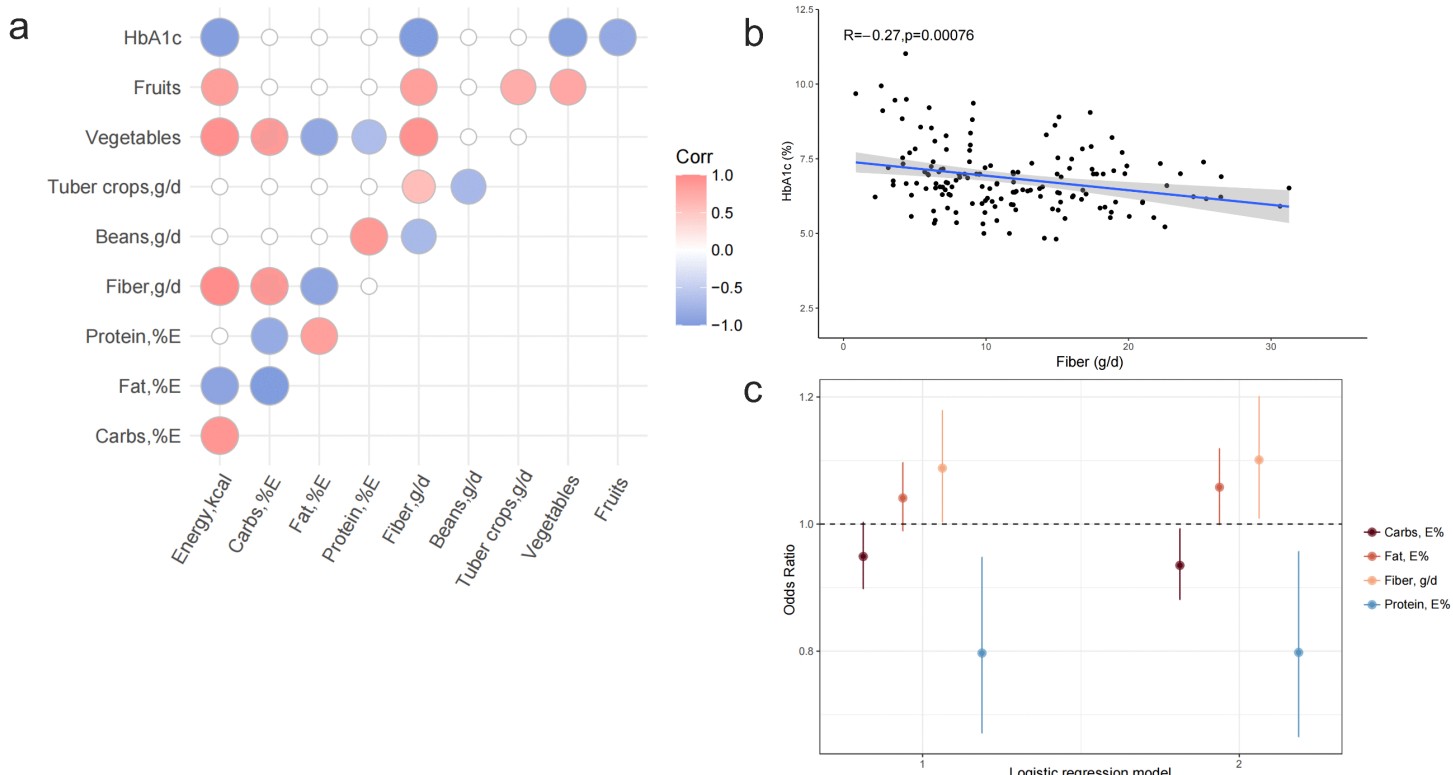

**Fig 2. Correlations between dietary nutrients and HbA1c. a.** Heat map illustrating the correlation between dietary nutrients and HbA1c; **b.** Scatter plots depicting the relationship between dietary fiber intake and HbA1c levels; **c.** Forest plot illustrating the correlation between dietary nutrients and outcomes in glycemic control.

SD (Est. = −0.83, p < 0.01, Table 4) and LAGE (Est. = −2.4, p < 0.01, Table 4). A modest but statistically significant inverse relationship was also observed for MAGE (Est. = −1.47, p = 0.04, Table 4). Additionally, a moderately significant positive correlation was found between the proportion of whole grains and TIR (Est. = 12.19, p = 0.01, Table 4).

### Meal-timing and glycemic variability: The greatest glycemic variability occurs at lunch compared to breakfast and dinner in T1D individuals

The fluctuation of postprandial glucose presented a significant challenge for individuals diagnosed with T1D, especially those who were encountering islet failure. To investigate the glycemic characteristics of T1D individuals, we analyzed

**Table 2. Associations between meal nutrient composition and HbA1c levels.**

| Meal composition | Crude model | | Model 1 | | Model 2 | |
|---|---|---|---|---|---|---|
| | β (95% CI) | p value | β (95% CI) | p value | β (95% CI) | p value |
| Carbs, E% | −0.16 (−0.18, −0.13) | 0.06 | 0.15 (0.12, 0.18) | 0.14 | 0.17 (0.14, 0.20) | 0.10 |
| Protein, E% | 0.15 (0.09, 0.22) | 0.06 | 0.27 (0.21, 0.34) | <0.01** | 0.23 (0.17, 0.30) | <0.01** |
| Fat, E% | 0.02 (−0.04, 0.05) | 0.79 | −0.18 (−0.21, −0.16) | 0.03* | −0.21 (−0.24, −0.18) | 0.02* |
| Fiber, g/d | −0.30 (−0.33, −0.27) | <0.01** | −0.31 (−0.34, −0.27) | <0.01** | −0.32 (−0.35, −0.28) | <0.01** |

Multivariate linear regression; *p < 0.05; **p < 0.01.

Model 1 adjusted for other macro-nutrients listed.

Model 2, as for Model 1, and adjusted for age, duration, BMI, total insulin dose, insulin regimen and physical activity.

**Table 3. Odds ratio of HbA1c≤6.5% based on meal nutrient composition.**

| Meal compositon | Model 1 | | | Model 2 | | |
|---|---|---|---|---|---|---|
| | Exp(B) | 95% CI | p value | Exp(B) | 95% CI | p value |
| Carbs, %E | 0.949 | 0.898~1.003 | 0.065 | 0.935 | 0.881~0.993 | 0.028* |
| Protein, %E | 0.797 | 0.671~0.948 | 0.010* | 0.798 | 0.665~0.957 | 0.015* |
| Fat, %E | 1.041 | 0.989~1.097 | 0.124 | 1.058 | 0.999~1.119 | 0.050 |
| Fiber, g/d | 1.088 | 1.003~1.179 | 0.042* | 1.101 | 1.009~1.201 | 0.031* |

Logistic regression; *p < 0.05; **p < 0.01.

Model 1 adjusted for other macronutrients listed.

Model 2, as for Model 1, and adjusted for age, duration, BMI, total insulin dose, insulin regimen and physical activity.

the postprandial blood glucose levels following three daily meals. In the mixed-effect model with post hoc Bonferroni test for multiple comparisons, we observed that postprandial blood glucose variability (SD, LAGE and MAGE) was highest at lunch compared to breakfast and dinner (1.53(0.85), p < 0.05; 5.44 (2.68), p < 0.05; 3.49(1.81), p < 0.05; Table 5). Then, we conducted a further analysis of the dietary characteristics of the three meals and discovered GI and GL were highest at lunch compared to breakfast and dinner (Table 5), and dietary fiber intake was greater during lunch compared to breakfast, but comparable to dinner (Table 5).

### Sources of carbohydrates and glycemic variability: A higher proportion of whole grains in carbohydrate sources is associated with reduced postprandial glucose variability

To further investigate the dietary factors influencing post-lunch blood glucose fluctuations, we performed MCA analysis. We found that in individuals with better glycemic control (TIR ≥ 70%, Peak BG < 10 mmol/L, and lower postprandial blood glucose AUC) (quadrant III, Fig 4), the proportion of whole grains in the carbohydrate sources was higher (≥40%). Furthermore, mix-effect model revealed a significant inverse relationship between dietary fiber intake and blood glucose variability (SD, LAGE and MAGE) (Est. = −0.06, p < 0.01; Est. = −0.19, p < 0.01; Est. = −0.12, p = 0.01, Table 6). Conversely, GI and GL exhibited a significant positive correlation with blood glucose fluctuations (Table 6). Furthermore, the proportion of whole grains in the carbohydrate source of lunch demonstrated a negative association with blood glucose variability (SD, LAGE and MAGE) (Est. = −0.45, p = 0.02; Est. = −1.52, p = 0.01; Est. = −1.39, p < 0.01, Table 6). Similarly, no significant correlation remained between CGM parameters and carbohydrate %E (Table 6) during lunch.

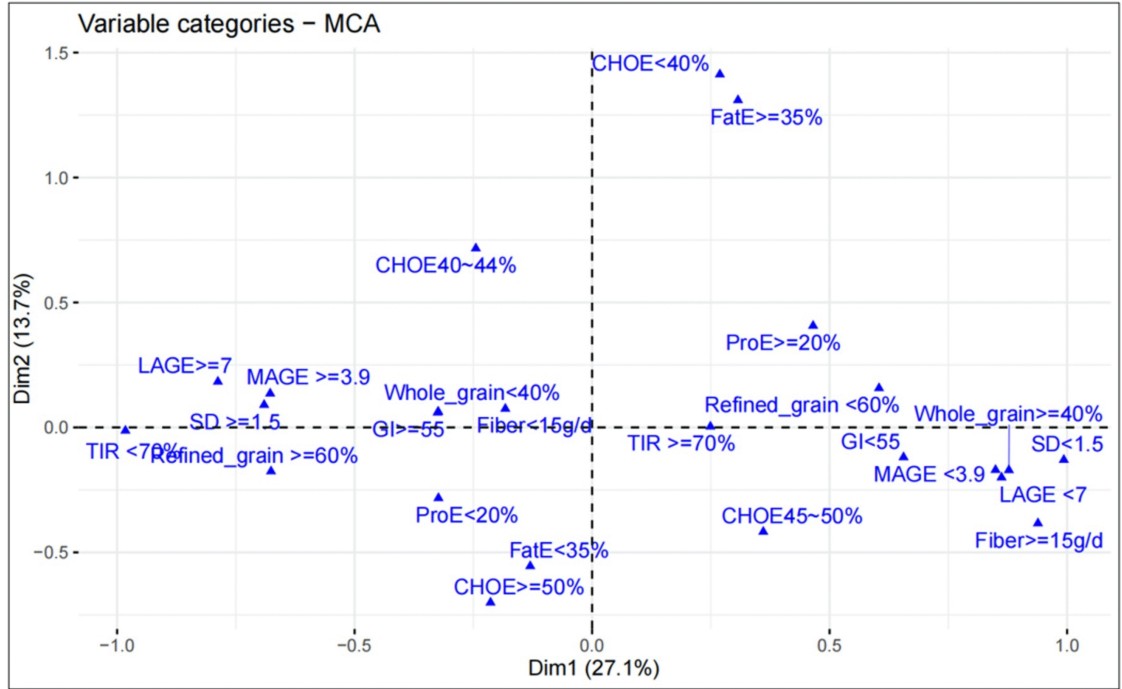

**Fig 3. Associations between dietary nutrients and glycemic control.** Results of the multiple correspondence analysis (MCA). CHOE<40%: % of total carbohydrate less than 40%; CHOE40～44%: between 40% and 44%; CHOE45～50: between 45% and 50%; CHOE≥50%: greater than 50%; FatE<35%: % of fat<35%; FatE≥35%: % of fat≥35%; ProE<20%: % of protein less than 20%; ProE≥20: greater than 20%; TIR≥70%: % time 3.9～10 mmol/L more than 70%; TIR<70%: % time 3.9～10 mmol/L less than 70%; Fiber<15g/d: total fiber intake less than 15g per day; Fiber≥15g/d: more than 15g per day; Whole_grain<40%: the proportion of whole grains in the carbohydrate source less than 40%; Whole_grain≥40%: greater than 40%; Refined_grain<60%: the proportion of refined grains in the carbohydrate source less than 60%; Refined_grain≥60%: greater than 60%. TIR, Time in range; SD, Standard Deviation; LAGE, Largest Amplitude of Glycemic Excursions; MAGE, Mean Amplitude of Glycemic Excursions; GI, Glycemic Index.

## Discussion

Individuals with T1D can enter a stage of extreme glycemic instability following islet failure. This phase is characterized by severe glycemic lability and recurrent hypoglycemia, which lead to profound glucose fluctuations and the imperative for lifelong, carefully managed insulin replacement therapy. [31]. In addition to the primary insulin treatment, dietary management plays a crucial role in controlling glycemic levels for individuals with T1D. Dietary guidelines for diabetes have shifted from strict carbohydrate restriction to a focus on carbohydrate quality, particularly emphasizing the intake of dietary fiber and prioritizing consideration of the GI and GL of foods [13]. Current studies have been primarily focused on the quality of carbohydrates in relation to T2D [32]. However, the research literature on the role of dietary carbohydrates in T1D is comparatively scant. Predominantly, such studies that do exist have concentrated on the implications of low-carbohydrate regimens on glycemic control in individuals with T1D [33, 34]. However, a carbohydrate-restricted or carbohydrate-free diet may not be universally viable or enduring in the long term. Additionally, there is a discrepancy in the results regarding the quality of carbohydrates, a study conducted in a large cohort of well-defined individuals with T1D in Finland [6], reported fiber intake was associated with lower mean blood glucose measurements. However, other investigations have failed to detect a substantial correlation between dietary fiber consumption and TIR [35]. Furthermore, research focusing on the relationship between carbohydrates and glycemic control in T1D is predominantly concentrated in European and American populations, with a relative paucity of corresponding data available for the Chinese demographic. Consequently, we aim to carry out a dietary investigation within the Chinese T1D cohort to examine the dietary preferences of these

**Table 4. Associations between meal nutrient composition and glycemic measures in CGM.**

| Meal composition | model | TIR | | SD | | LAGE | | MAGE | |
|---|---|---|---|---|---|---|---|---|---|
| | | β (95% CI) | p value | β (95% CI) | p value | β (95% CI) | p value | β (95% CI) | p value |
| Fat, %E | 1 | 0.12 (−0.01, 0.24) | 0.06 | −0.15 (−0.29, −0.02) | 0.03* | −0.12 (−0.25, 0.01) | 0.07 | −0.08 (−0.22, 0.07) | 0.31 |
| | 2 | 0.12 (0.001, 0.24) | 0.04 | −0.15 (−0.29, −0.02) | 0.02* | −0.10 (−0.24, 0.03) | 0.12 | −0.07 (−0.22, 0.08) | 0.36 |
| Protein, %E | 1 | 0.05 (−0.06, 0.15) | 0.37 | −0.08 (−0.21, 0.02) | 0.12 | −0.09 (−0.20, 0.02) | 0.11 | −0.09 (−0.22, 0.04) | 0.19 |
| | 2 | 0.05 (−0.05, 0.15) | 0.33 | −0.09 (−0.19, 0.04) | 0.19 | −0.07 (−0.19, 0.04) | 0.20 | −0.08 (−0.22, 0.05) | 0.21 |
| Carbs, %E | 1 | 0.08 (−0.08, 0.24) | 0.31 | 0.01 (−0.16, 0.19) | 0.88 | −0.03 (−0.20, 0.14) | 0.75 | −0.11 (−0.31, 0.09) | 0.29 |
| | 2 | 0.09 (−0.07, 0.24) | 0.27 | 0.005 (−0.17, 0.18) | 0.96 | −0.04 (−0.21, 0.13) | 0.63 | −0.11 (−0.31, 0.08) | 0.25 |
| Fiber, g/d | 1 | 0.12 (−0.01, 0.24) | 0.07 | −0.263 (−0.40, −0.13) | <0.01** | −0.25 (−0.38, −0.12) | <0.01** | −0.22 (−0.37, −0.07) | <0.01** |
| | 2 | 0.12 (−0.01, 0.24) | 0.06 | −0.26 (−0.40, −0.13) | <0.01** | −0.25 (−0.39, −0.12) | <0.01** | −0.23 (−0.38, −0.08) | <0.01** |
| Glycemic Index, % | 1 | −0.04 (−0.15, 0.07) | 0.47 | 0.14 (0.02, 0.26) | 0.03* | 0.10 (−0.02, 0.21) | 0.11 | 0.13 (−0.01, 0.27) | 0.06 |
| | 2 | −0.03 (−0.14, 0.07) | 0.55 | 0.14 (0.02, 0.26) | 0.03* | 0.10 (−0.02, 0.21) | 0.11 | 0.11 (−0.03, 0.25) | 0.13 |
| Glycemic Load, units | 1 | −0.07 (−0.25, 0.10) | 0.39 | 0.26 (0.07, 0.45) | 0.01* | 0.14 (−0.04, 0.34) | 0.13 | 0.09 (−0.12, 0.30) | 0.42 |
| | 2 | −0.06 (−0.23, 0.11) | 0.50 | 0.23 (0.04, 0.43) | 0.02* | 0.10 (−0.08, 0.29) | 0.27 | 0.01 (−0.21, 0.23) | 0.99 |
| Whole-grain, %CHO | 1 | 0.16 (0.04, 0.28) | 0.01* | −0.23 (−0.36, −0.10) | <0.01** | −0.18 (−0.31, −0.05) | 0.01* | −0.17 (−0.32, −0.03) | 0.02* |
| | 2 | 0.15 (0.03, 0.27) | 0.01* | −0.24 (−0.38, −0.11) | <0.01** | −0.19 (−0.32, −0.06) | <0.01** | −0.15 (−0.30, −0.01) | 0.04* |
| Refined-grain, %CHO | 1 | −0.17 (−0.29, −0.05) | 0.01* | 0.24 (0.11, 0.37) | <0.01** | 0.21 (0.08, 0.34) | <0.01** | 0.18 (0.03, 0.33) | 0.02* |
| | 2 | −0.16 (−0.28, −0.05) | 0.01* | 0.23 (0.10, 0.37) | <0.01** | 0.20 (0.07, 0.33) | <0.01** | 0.17 (0.02, 0.32) | 0.02* |

* $p < 0.05$; **$p < 0.01$; mix-effect model.

Model 1 adjusted for other macro-nutrients listed.

Model 2, as for Model 1, and adjusted for age, duration, BMI, total insulin dose, Insulin regimen and physical activity.

TIR, Time in range; SD, Standard Deviation; LAGE, Largest Amplitude of Glycemic Excursions; MAGE, Mean Amplitude of Glycemic Excursions. Whole-grain, %CHO, the proportion of refined grains in the carbohydrate source; Refined-grain, %CHO, the proportion of refined grains in the carbohydrate source.

individuals and clarify the relationship between the quality of carbohydrates consumed and glycemic management. It is expected that without changing the proportion of carbohydrates, the quality of carbohydrates can be optimized, with the goal of improving glycemic control in individuals with T1D.

Our research is pioneering in examining the correlation between diet and real-time dynamic blood glucose levels through the integration of dietary records in individuals with T1D. Despite being an observational study, the findings are both accurate and reliable. Our findings indicate that increased intake of dietary fiber and consumption of low GI and GL

**Table 5. The glycemic profiles and nutrient distribution in the three meals of T1D.**

| Variables | Breakfast | Lunch | Dinner |
|---|---|---|---|
|  | (n = 166) | (n = 188) | (n = 184) |
| **Glycemic measures in CGM** | | | |
| TIR | 81.9 (27.0) | 80.7 (23.8) | 81.0 (23.9) |
| TAR | 14.8 (26.6) | 15.3 (23.5) | 14.5 (23.7) |
| TBR | 3.3 (9.1) | 4.0 (9.1) | 4.5 (10.0) |
| SD | 1.25 (0.821) | 1.53 (0.850)* † | 1.33 (0.790) |
| MAGE | 2.21 (1.21) | 3.49 (1.81)* † | 2.92 (1.69)* |
| LAGE | 4.22 (2.39) | 5.44 (2.68)* † | 4.79 (2.47)* |
| **Meal composition** | | | |
| Fat, %E | 34.2 (13) | 32.5 (14.6) | 30.5 (14.5)* |
| Carbs, %E | 48.7 (16.1) | 50.3 (16.5) | 50.8 (17.6) |
| Protein, %E | 18.7 (5.4) | 21.3 (10.8)* | 20.3 (7.3) |
| Fiber, g/d | 2.1 (2.2) | 4.0 (3.4)* | 3.8 (3.2)* |
| Glycemic index, % | 52.9 (16.1) | 61.7 (14.1)* † | 57.2 (16.2)* |
| Glycemic load, units | 27.9 (18.3) | 46.7 (23.5)* † | 40.1 (21.3)* |
| Whole-grain, %CHO | 24.1 (33.6) | 23.7 (31.9) | 23.0 (31.2) |
| Refined-grain, %CHO | 43.9 (39.5) | 52.4 (36.9) | 51.5 (37.1) |

*Data are presented as means (SD).

*p < 0.05 vs breakfast; †p < 0.05 vs dinner (mixed-effect model with post hoc Bonferroni test for multiple comparisons); n: number of meals.

foods are linked to improved glycemic control in T1D individuals, and incorporate a higher proportion of whole grains in carbohydrate sources may be associated with reduced postprandial glycemic fluctuations.

The Chinese Diabetes Medical Nutrition Treatment Guidelines (2022) emphasize that a high-fiber diet (25-36g/d) can lead to improved glycemic control and reduced all-cause mortality [19]. However, our dietary survey revealed that the median daily intake of dietary fiber in our study population was only 10.32g/d, significantly below the recommended amount. Additionally, our findings revealed that refined grains made up nearly 90% of staple foods, while whole grains constituted a small proportion. This contradicts China's medical nutrition treatment guidelines, which recommend replacing some refined grains with whole grain carbohydrates to help control blood glucose, TG levels, and weight.

Our findings demonstrated a significant inverse association between dietary fiber intake and HbA1c levels, indicating improved long-term glycemic control, which was in line with earlier research results [6,36]. The potential mechanism underlying the beneficial impact of dietary fiber on blood glucose may encompass the following aspects. Dietary fiber can modulate gut microbiota composition and enhance its diversity. Additionally, dietary fiber may impact the glycemic response to a meal by increasing intraluminal viscosity in the small intestine and slowing carbohydrate absorption [13]. Dietary fiber may also promote insulin sensitivity through effects on the gut microbiome, production of short-chain fatty acids, and other actions [37]. In our study, dietary fiber intake demonstrated a significant positive association with achieving glycemic targets (OR = 1.101, 95% CI: 1.009–1.201). Although the effect size may seem limited, its public health relevance should not be underestimated. These findings are consistent with the known physiological effects of dietary fiber, including delayed gastric emptying and attenuation of postprandial glucose excursions. Crucially, the association between dietary fiber and glycemic outcomes (OR=1.101) must be contextualized by the low median fiber intake (10.32g/d) in our cohort. Therefore, a linear extrapolation of this effect to the recommended daily intake (25-36g/d) is not appropriate, as the dose-response relationship may be non-linear and exhibit diminishing returns at higher levels. In addition, as with any observational study, we cannot rule out the possibility of residual confounding due to unmeasured or imperfectly

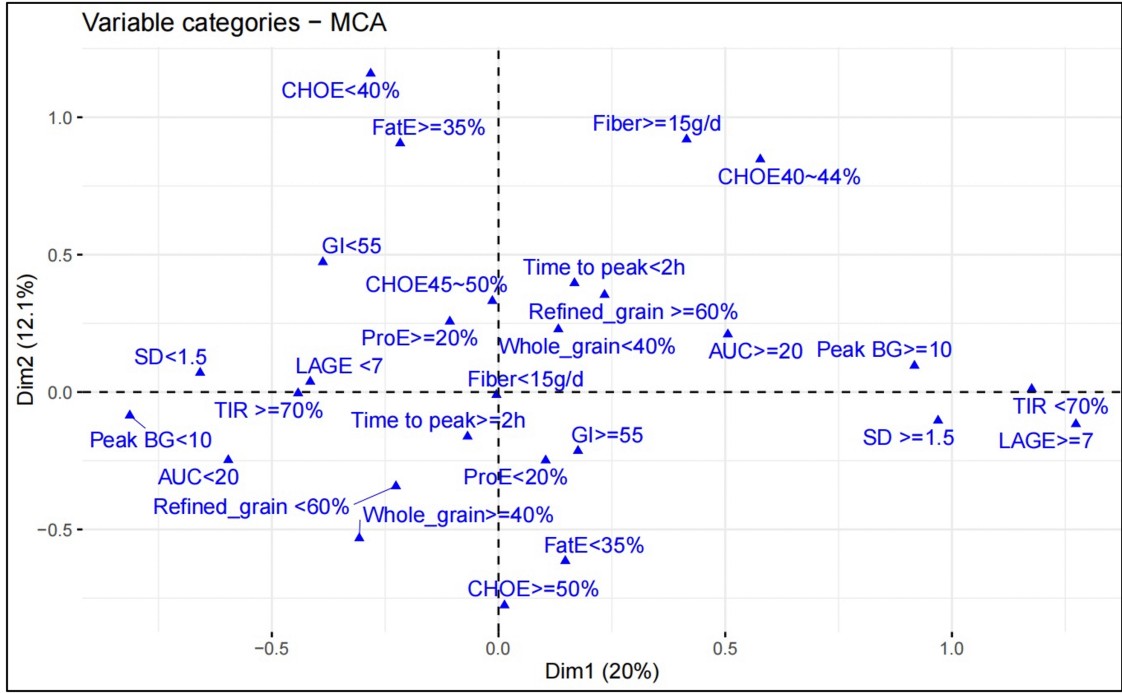

**Fig 4. Correlations between dietary factors and postprandial glycemic control after lunch.** Results of the multiple correspondence analysis (MCA). Peak BG < 10: Peak blood glucose less than 10 mmol/L after lunch; Peak BG ≥ 10: greater than 10 mmol/L after lunch; AUC < 20: Area under the curve of blood glucose curves at 0 min, 30 min, 60 min, 120 min and 180 min after lunch less than 20 mmol/L·h; AUC ≥ 20: greater than 20 mmol/L·h; Time to peak < 2h: The time to reach peak blood glucose after lunch less than 2 hours; Time to peak ≥ 2h: greater than 2 hours.

measured factors, such as overall dietary quality, socioeconomic status, or other healthy lifestyle behaviors associated with higher fiber intake.

The 3-day dietary records combined with CGM revealed a significant negative correlation between dietary fiber intake and glucose fluctuation indicators (SD, MAGE, LAGE). Subsequent analysis of postprandial blood glucose levels following three meals aimed to investigate the blood glucose characteristics of individuals with T1D. After adjusting for disease duration and insulin dose, it was observed that lunch resulted in the largest glucose fluctuation compared to breakfast and dinner. Furthermore, mixed linear regression analysis demonstrated a significant positive correlation between GI and GL with postprandial blood glucose fluctuation indicators in individuals with T1D. This indicates that foods with high GI and GL levels may be the primary factor contributing to postprandial blood glucose variability. It is worth noting that our study suggested a potential differential effect of dietary protein in T1D: habitual intake (FFQ) was associated with long-term glycaemic control (HbA1c), while short-term intake (3-day records) showed no link to acute variability. This aligns with evidence that large acute protein loads can induce delayed hyperglycaemia via gluconeogenesis and glucagon secretion [38,39], yet the effect is multifactorial, influenced by protein type, co-ingested nutrients, and insulin dosing [40–42]. This contrasts with T2D, where high-protein diets often improve glycaemia [43]. Thus, the glycaemic impact of protein in T1D appears complex and distinct from T2D, warranting further investigation to inform tailored dietary advice. However, our findings are constrained by the small sample size, recall bias inherent to FFQs and the cross-sectional design, precluding causal inference. Prospective cohort studies and rigorously controlled randomized trials with refined methodological approaches are required to delineate the true relationship between dietary protein and glycaemic outcomes in T1D.

**Table 6. The associations between dietary factors and postprandial glycemic variability at lunch.**

| Meal composition | TIR | | SD | | LAGE | | MAGE | |
|---|---|---|---|---|---|---|---|---|
| | β (95% CI) | p value | β (95% CI) | p value | β (95% CI) | p value | β (95% CI) | p value |
| Fat, %E | 0.14 (−0.01, 0.28) | 0.06 | −0.13 (−0.27, 0.01) | 0.07 | −0.12 (−0.26, 0.01) | 0.08 | −0.14 (−0.31, 0.02) | 0.10 |
| Protein, %E | 0.05 (−0.07, 0.18) | 0.41 | −0.06 (−0.20, 0.07) | 0.34 | −0.06 (−0.19, 0.07) | 0.40 | −0.10 (−0.26, 0.05) | 0.21 |
| Carbs, %E | −0.08 (−0.22, 0.05) | 0.24 | 0.06 (−0.08, 0.20) | 0.41 | 0.04 (−0.10, 0.18) | 0.55 | 0.03 (−0.14, 0.21) | 0.74 |
| Fiber, g/d | 0.05 (−0.09, 0.19) | 0.50 | −0.23 (−0.38, −0.09) | <0.01** | −0.25 (−0.39, −0.11) | <0.01** | −0.23 (−0.39, −0.06) | 0.01* |
| Glycemic Index, % | −0.09 (−0.24, 0.05) | 0.21 | 0.22 (0.07, 0.37) | <0.01** | 0.23 (0.08, 0.37) | <0.01** | 0.22 (0.06, 0.39) | 0.01* |
| Glycemic Load, units | −0.16 (−0.34, 0.03) | 0.10 | 0.26 (0.07, 0.44) | 0.01* | 0.25 (0.06, 0.43) | 0.01* | 0.30 (0.10, 0.49) | <0.01** |
| Whole-grain, %Carbs | 0.06 (−0.08, 0.20) | 0.39 | −0.17 (−0.31, −0.03) | 0.02* | −0.18 (−0.32, −0.04) | 0.01* | −0.24 (−0.41, −0.09) | <0.01** |
| Refined-grain, %Carbs | −0.07 (−0.21, 0.07) | 0.35 | 0.12 (−0.02, 0.27) | 0.09 | 0.14 (−0.01, 0.29) | 0.05 | 0.28 (0.12, 0.44) | <0.01** |

* $p < 0.05$; ** $p < 0.01$.

Mix-effect model.

Considering the participant's ID as random effect and day as a nested random effect.

Adjusted for age, duration, BMI, total insulin dose, Insulin regimen, physical activity and other macro-nutrients listed.

This study clinically demonstrates significant associations between dietary fiber intake and both long-term glycemic control and HbA1c target attainment (≤6.5%) in T1D individuals. Additionally, our findings indicate that individuals with T1D experience significant post-lunch blood glucose fluctuations, commonly known as the "dusk phenomenon" [44]. Therefore, choosing foods with low GI and GL such as increasing the proportion of whole grains in carbohydrate sources during this time may help mitigate these fluctuations. This provides valuable guidance for the daily dietary choices of individuals with T1D. For individuals with T1D, higher consumption of whole grains as a primary carbohydrate source, particularly during lunch, is associated with attenuated postprandial glycemic excursions when a low-carbohydrate diet pattern is not required. Glycemic variability observed during the lunch period is an intriguing result that merits deeper exploration. Further studies are warranted to elucidate the underlying mechanisms of this phenomenon, particularly the roles of lunch-specific meal composition, meal timing, circadian rhythms, and physical activity patterns.

An intriguing finding was the greatest glycemic variability at lunch, despite its highest fiber intake. This paradox underscores the multifactorial nature of glycemic control. Several factors may explain this: 1) the postprandial response is shaped by more than fiber, as co-ingested protein and fat can delay gastric emptying and prolong glycemic fluctuations [45]; 2) diurnal insulin resistance in the afternoon may blunt fiber's benefit; 3) the concurrently high meal GI/GL likely offset the high fiber's advantage by presenting a greater overall carbohydrate challenge; and 4) behavioral factors like bolus timing accuracy, particularly Automated Insulin Delivery (AID) systems [46], may differ at lunch. Consequently, nutritional advice should consider the overall meal context and concomitant behaviors, rather than focusing on single nutrients.

Our study also has some limitations. First, it is a cross-sectional study, and all results are based on association analysis. Compared to randomized controlled clinical trials, the level of evidence is relatively weak. Second, our sample size was relatively small in comparison to other similar studies [6,47]. The sample size reduces power to detect smaller effects and limits subgroup analyses. However, most previous studies primarily used self-blood glucose monitoring or

HbA1c to evaluate blood glucose control. Apart from this, in a recent study of a large sample of T1D individuals, dietary records were used to assess the impact of dietary factors while glycemic control was evaluated using CGM; However, the two assessments were not synchronized [6,47]. Third, dietary data reliance on self-report may introduce measurement bias, though we mitigated this through standardized collection tools. Our advantage lies in the fact that individuals wore CGM while recording their diet, allowing us to obtain rich and detailed blood glucose data which is more suitable for studying the relationship between diet and blood glucose in T1D individuals. In addition, there is a certain degree of recall bias in the FFQ; However, the subsequent 3-day diet recording is currently considered the gold standard for evaluating dietary composition. The combination of these two methods can enhance the reliability of our conclusions. Furthermore, all dietary records are completed under the guidance of a professional dietitian and meticulously checked to minimize errors. However, although we implemented standardized training, recall bias and portion estimation errors remain unavoidable. Future studies with larger sample sizes or well-designed randomised controlled trials are still needed to further clarify the relationship between carbohydrate quality and glycaemic control in individuals with T1D. Besides this, future studies should aim to integrate objective biomarkers of dietary intake. For instance, plasma or urinary alkylresorcinols, which are specific markers of whole-grain wheat and rye consumption, can be used to complement and validate self-reported data. This approach would reduce misclassification and improve the accuracy of nutritional assessments in relation to glycemic outcomes. Notably, owing to potential variations in genetic background, diet, lifestyle, and healthcare systems, the findings from our Chinese cohort should be extrapolated with caution to other ethnic populations. Most importantly, as an observational study, it is susceptible to unmeasured confounding. Factors such as carbohydrate counting accuracy, hypoglycemia management behaviors, and socioeconomic status were not measured in this analysis. These factors could influence both dietary choices and glycemic control (HbA1c), and thus may partly account for the observed associations. For instance, individuals with higher health literacy or socioeconomic status might simultaneously have better access to high-fiber foods and achieve better glycemic management. Future studies that meticulously collect data on these potential confounders are needed to confirm the magnitude and independence of these dietary effects. Finally, the statistical power of this study was limited by the sample size. Although it was adequate for detecting moderate-to-large effects, the study was likely underpowered to identify smaller, yet potentially important, associations. For instance, the non-significant finding for the association between protein energy proportion at lunch and postprandial glycemic variability should be interpreted with caution, as it may reflect a Type II error rather than conclusive evidence for the absence of an effect.

Despite the observed benefits, translating these findings into practice faces several challenges specific to the T1D population. A primary concern is the practical difficulty of accurately estimating the carbohydrate content of whole grain foods for insulin dosing, as their high fiber content can affect glycemic response and necessitate individualized insulin dose adjustments. Furthermore, initial gastrointestinal discomfort, suboptimal palatability, limited availability, and deeply ingrained dietary habits collectively pose significant barriers to the adherence and acceptance of whole-grain products. Therefore, any recommendation to increase whole grain intake should be accompanied by structured nutritional education, close glucose monitoring, and collaboration between individuals and dietitians to tailor changes to individual lifestyles and insulin regimens.

## Conclusions

In summary, for individuals with T1D, it is not necessary to adopt a low-carbohydrate diet pattern. Evidence from this study indicates that elevated dietary fiber intake is significantly associated with enhanced long-term glycemic control in individuals with T1D. Notably, our findings reveal that preferential consumption of whole grains as the predominant carbohydrate source, with particular emphasis on lunchtime intake, is correlated with reduced postprandial glucose variability. As these results are exploratory, they should be interpreted as hypothesis-generating and require validation in independent cohorts.

## Supporting information

**S1 File.** Supporting information. Contains: S1 Fig. Distribution of the daily macro-nutrient proportions derived from the total energy intake; S2 Fig. Box-plot illustrating the correlation between carbohydrate (%E) intake and postprandial blood glucose AUC; S1 Table. Baseline characteristics of individuals with T1D in FFQ cohort; S2 Table. Characteristics of nutrient intake and indicators for glycemic control derived from CGM in 3-day food records; S3 Table. Missing values per variable (FFQ cohort, n = 155); S4 Table. Missing values per variable (3d24h dietary record cohort, number of days = 188); S5 Table. Missing values per variable (3d24h dietary record cohort, number of lunch = 188).
(DOCX)

**S2 File. Original datasets.**
(ZIP)

## Acknowledgments

We are indebted to all the people who kindly participated in this study.

## Author contributions

**Conceptualization:** Yanjun Jin, Jianling Bai, Tao Yang, Yong Gu, Hechun Liu.

**Data curation:** Lingling Bian, Chun Yang, Hong Wang, Min Zhu, Jingjing Xu.

**Formal analysis:** Lingling Bian, Hechun Liu.

**Funding acquisition:** Tao Yang, Yong Gu, Hechun Liu.

**Investigation:** Hong Wang, Mei Zhang, Yanmei Liu.

**Project administration:** Mei Zhang.

**Resources:** Hong Wang, Min Zhu, Jingjing Xu.

**Software:** Yanjun Jin.

**Supervision:** Jianling Bai, Yanmei Liu, Tao Yang, Yong Gu.

**Validation:** Lingling Bian, Jianling Bai, Yong Gu, Hechun Liu.

**Visualization:** Chun Yang, Yanjun Jin.

**Writing – original draft:** Lingling Bian.

**Writing – review & editing:** Lingling Bian, Chun Yang.

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
