## [Decision Letter · Decision Letter 0]

26 Jun 2025

Dear Dr. Liu,

Thank you for submitting your manuscript to PLOS ONE. After careful consideration, we feel that it has merit but does not fully meet PLOS ONE’s publication criteria as it currently stands. Therefore, we invite you to submit a revised version of the manuscript that addresses the points raised during the review process.

We look forward to receiving your revised manuscript.

Kind regards,

Gbolahan Deji Olatunji, M.D, MPH

Academic Editor

PLOS ONE

Journal Requirements:

2. Thank you for stating the following financial disclosure: [This research was funded by the Noncommunicable Chronic Diseases-National Science and Technology Major Project (NO. 2023ZD0507400, 2023ZD0507401, 2023ZD0507403), National Natural Science Foundation of Jiangsu Province (NO. BK20220708) and National Natural Science Foundation of China (NO. 82170837, 82230028, 82404271).].

Reviewers' comments:

Reviewer's Responses to Questions

**Comments to the Author**

1. Is the manuscript technically sound, and do the data support the conclusions?

Reviewer #1: Partly

Reviewer #2: Partly

Reviewer #3: Yes

2. Has the statistical analysis been performed appropriately and rigorously?

Reviewer #1: No

Reviewer #2: I Don't Know

Reviewer #3: Yes

3. Have the authors made all data underlying the findings in their manuscript fully available?

Reviewer #1: No

Reviewer #2: No

Reviewer #3: No

4. Is the manuscript presented in an intelligible fashion and written in standard English?

Reviewer #1: Yes

Reviewer #2: Yes

Reviewer #3: Yes

Reviewer #1: Major Concerns:

The cross-sectional design cannot support the causal language used throughout the manuscript. Terms like "beneficial," "improving," and recommendations to "increase consumption" are inappropriate for observational data. The title claiming "Improved Carbohydrate Quality" overstates what can be concluded from cross-sectional associations.

Critical statistical information is missing from the analysis. The authors report no effect sizes for key findings, provide no confidence intervals for primary outcomes, and include no power calculations or sample size justifications. Multiple testing corrections were not applied despite numerous dietary variable comparisons.

The methodological rigor is insufficient for publication standards. The CGM monitoring period of only 3 days falls below current standards requiring ≥14 days for reliable metrics. Missing data handling procedures are not described, and LightGBM cross-validation methods are inadequately documented.

Data Availability Issues:

The manuscript does not comply with PLOS data availability requirements. Despite claiming "all relevant data are within the manuscript," only summary statistics are provided. Individual-level data, raw CGM measurements, and machine learning datasets required by PLOS policy are missing.

Technical Problems:

The sample size of 65 participants for CGM analysis is insufficient for robust meal-specific conclusions. Selection bias from clinic-based recruitment is inadequately addressed. There is temporal mismatch between dietary assessment periods and CGM monitoring. Key confounders including medication adherence and socioeconomic factors are not controlled.

Required Revisions:

All conclusions must be reframed using associational rather than causal language. Effect sizes with confidence intervals must be reported for all key findings. Design limitations must be acknowledged prominently in the abstract and conclusions. Clinical recommendations inappropriate for observational data must be removed. Complete individual-level datasets must be provided per PLOS policy requirements.

The research question is clinically relevant and the statistical approaches are sophisticated. However, the fundamental mismatch between study design capabilities and conclusions drawn prevents acceptance. Major revisions addressing these core methodological and reporting issues are required before this work can meet publication standards.

Reviewer #2: 1. The current title does not fully capture the scope and analytical depth of the study. While it accurately reflects the general finding, it overlooks key aspects such as the study design (cross-sectional), the use of both FFQ and 3-day meal-based dietary records, and the incorporation of CGM-derived glycemic variability metrics analyzed through mixed-effects and machine learning models.

I recommend revising the title to better reflect these methodological components and provide a clearer picture of the study at a glance. A more appropriate and informative title might be:

“Association Between Carbohydrate Quality and Glycemic Control in Individuals with Type 1 Diabetes: A Cross-Sectional and Meal-Based Analysis”

Or

“Carbohydrate Quality and Glycemic Outcomes in Type 1 Diabetes: Evidence from Cross-Sectional and Meal-Level Data Analyses”

2. The association between higher dietary protein intake and elevated HbA1c levels (Tables 2 and 3) contrasts with existing literature that suggests protein intake is typically neutral or beneficial for glycemic control in T1D. The discussion should address possible explanations for this discrepancy, including residual confounding, differences in protein source (e.g., animal vs. plant), dietary context, or reverse causality (e.g., individuals with poor control increasing protein intake following clinical advice).

3. The interpretation of the MCA results suggests an association between high-quality carbohydrate intake and better glycemic control. It should be emphasized that these findings are observational and do not imply causality due to the cross-sectional design. Clarify whether the dietary variables independently contributed to quadrant clustering or whether other unmeasured factors may have influenced the patterns.

4. Misclassification bias should be acknowledged as a limitation. Despite using validated tools, self-reported dietary data—especially regarding portion sizes, meal timing, and food type classification—are subject to error and may affect the observed associations.

5. There is no table in the manuscript or supplementary materials that clearly presents the demographic and clinical characteristics of the study participants (e.g., age range, gender distribution, BMI, diabetes duration, insulin regimen). This information is essential for evaluating the generalizability and context of the findings. Please include a table in the main text or supplementary materials summarizing these baseline characteristics.

Reviewer #3: I commend the authors' work in evaluating the association between improved carbohydrate quality and better glycemic control in individuals with Tye 1 Diabetes.

Please find my comments below:

-line 29: Please replace "established" with "used" to describe the utilization of LightGBM model in the analysis

-line 77: Please replace "dietary intake" with "the quality of carbohydrates in the diet" which better describes the aim of the study

-line 87: Exclusion criteria: Did the authors consider hypothyroidism specifically, given the association between hypothyroidism and Type 1 Diabetes?

-lines 100 - 101: "Subsequently, a total of 65 patients maintained a 3-day food record..."

Are there specific criteria for selecting the 65 participants who maintained a 3-day food record and wore the CGM? Please include the method of section in the manuscript.

-line 101: mentioned that these 65 participants wore the CGM:

How long were participants expected to wear the CGM?

Did all participants wear the CGM for the same length of time?

Please include this information in the manuscript.

-lines 119-124: described various analysis performed.

Was there additional analysis to test for multicollinearity among the independent variables?

**Do you want your identity to be public for this peer review?** For information about this choice, including consent withdrawal, please see our Privacy Policy

Reviewer #1: **Yes:** ELOHOR OBOREVWORI

Reviewer #2: No

Reviewer #3: No

---

## [Author Response · Author response to Decision Letter 1]

7 Aug 2025

Response to comments

We sincerely thank the editor and all reviewers for their valuable feedback that we have used to improve the quality of our manuscript. The reviewers’ comments are laid out below in italicized font and specific concerns have been numbered. Our response is given in red font and changes to the manuscript are highlighted in yellow. The highlighted modified sections in the text correspond to the line numbers of the Revised Manuscript (clean copy) version.

Reviewer #1:

Required Revisions:

1.All conclusions must be reframed using associational rather than causal language.

Response 1: Following the reviewer's suggestions, we have implemented several revisions in both the abstract and main text, as detailed below:

In the abstract section, we have replaced the original statement “Increasing the consumption of dietary fiber is beneficial for achieving long-term glycemic control. Additionally, increasing the intake of whole grains, especially at lunch, as a source of high-quality carbohydrates, is helpful in reducing glycemic fluctuations, which is of crucial significance for reducing the risk of long-term complications in T1D patients.” with “From a long-term glycemic control perspective, higher dietary fiber intake appears to be associated with improved HbA1c levels, while in terms of short-term glycemic variability, increased whole grain consumption is associated with reduced glucose fluctuations. However, as a cross-sectional analysis, these findings represent observational associations rather than causal evidence. Further validation through prospective cohorts and randomized trials is needed to assess clinical applicability.” (Line 38-43) to present a more balanced conclusion.

In the result session, we have replaced the original statement “Increasing the proportion of whole grains in the source of carbohydrates is beneficial for reducing variations in postprandial blood glucose levels” with “A higher proportion of whole grains in carbohydrate sources is associated with reduced postprandial glucose variability”. (Line 251-252)

In the discussion section, we have replaced the original statement “and incorporate a higher proportion of whole grains in carbohydrate sources may be beneficial for reducing postprandial glycemic fluctuations, offering potential guidance for dietary choices in this population” with “and incorporate a higher proportion of whole grains in carbohydrate sources may be associated with reduced postprandial glycemic fluctuations” (Line 325-326), and we have replaced the original statement “The clinical significance of this study lies in the discovery that dietary fiber intake is beneficial for long-term glycemic control and contributes to blood glucose compliance (HbA1c≤6.5%) in T1D patients” with “This study clinically demonstrates significant associations between dietary fiber intake and both long-term glycemic control and HbA1c target attainment (≤6.5%) in T1D patients.” (Line 376-377) , and have replaced the original statement “For patients with T1D, increasing the consumption of whole grains, particularly at lunch, as a source of high-quality carbohydrates, is advantageous for reducing post-meal glycemic fluctuations when a low-carbohydrate diet pattern is not required” with “For patients with T1D, higher consumption of whole grains as a primary carbohydrate source - particularly during lunch - is associated with attenuated postprandial glycemic excursions when a low-carbohydrate diet pattern is not required.” (Line 382-384)

In the conclusion section, we have replaced the original statement “Increasing the consumption of dietary fiber is beneficial for achieving long-term glycemic control. Additionally, increasing the intake of whole grains, especially at lunch, as a source of high-quality carbohydrates, is helpful in reducing post-meal glycemic fluctuations, which is of crucial significance for reducing the risk of long-term complications in T1D patients” with “Evidence from this study indicates that elevated dietary fiber intake is significantly associated with enhanced long-term glycemic control in individuals with T1D. Notably, our findings reveal that preferential consumption of whole grains as the predominant carbohydrate source, with particular emphasis on lunchtime intake, is correlated with reduced postprandial glucose variability.” (Line 408-411)

2.Effect sizes with confidence intervals must be reported for all key findings.

Response 2: Following the reviewer's suggestions, as shown in Table 2, Table 4 and Table 6, we have reported the standardized regression coefficients with 95% confidence intervals to facilitate effect size comparison. In the revised table, we replaced the unstandardized raw regression coefficients (Est.) with β values.

3.Design limitations must be acknowledged prominently in the abstract and conclusions.

Response 3: We sincerely appreciate the reviewers’ insightful comments regarding the study design. We fully acknowledge that as a cross-sectional study, our research provides a lower level of evidence compared to randomized controlled trials (RCTs), which we have now explicitly addressed in both the abstract, discussion and conclusion sections:

Abstract:

Added the following limitation statement:

“However, as a cross-sectional analysis, these findings represent observational associations rather than causal evidence. Further validation through prospective cohorts and randomized trials is needed to assess clinical applicability.” ( line 40-42)

Discussion:

“Our study also has some limitations. First, it is a cross-sectional study, and all results are based on association analysis. Compared to randomized controlled clinical trials, the level of evidence is relatively weak. Second, our sample size was relatively small in comparison to other similar studies. The sample size reduces power to detect smaller effects and limits subgroup analyses. However, most previous studies primarily used self-blood glucose monitoring or HbA1c to evaluate blood glucose control. Apart from this, in a recent study of a large sample of T1D patients, dietary records were used to assess the impact of dietary factors while glycemic control was evaluated using CGM; However, the two assessments were not synchronized. Third, dietary data reliance on self-report may introduce measurement bias, though we mitigated this through standardized collection tools.” ( line 385-394 )

4.Clinical recommendations inappropriate for observational data must be removed.

Response 4: We are grateful to the reviewers for this valuable comment. Indeed, our cross-sectional design limits the ability to formulate clinical guidelines. Accordingly, we have implemented the following modifications:

In the abstract, we have removed the clinical recommendation “Increasing the consumption of dietary fiber is beneficial for achieving long-term glycemic control. Additionally, increasing the intake of whole grains, especially at lunch, as a source of high-quality carbohydrates, is helpful in reducing glycemic fluctuations, which is of crucial significance for reducing the risk of long-term complications in T1D patients.” and replaced it with statement “From a long-term glycemic control perspective, higher dietary fiber intake appears to be associated with improved HbA1c levels, while in terms of short-term glycemic variability, increased whole grain consumption is associated with reduced glucose fluctuations. However, as a cross-sectional analysis, these findings represent observational associations rather than causal evidence. Further validation through prospective cohorts and randomized trials is needed to assess clinical applicability.” ( line 38-43 ) to maintain appropriate focus on our study findings rather than clinical guidance.

In the Discussion section, we have replaced the original statement “For patients with T1D, increasing the consumption of whole grains, particularly at lunch, as a source of high-quality carbohydrates, is advantageous for reducing post-meal glycemic fluctuations when a low-carbohydrate diet pattern is not required” with “For patients with T1D, higher consumption of whole grains as a primary carbohydrate source - particularly during lunch - is associated with attenuated postprandial glycemic excursions when a low-carbohydrate diet pattern is not required.” (Line 382-384).

In the conclusion section, we have replaced the original statement “Increasing the consumption of dietary fiber is beneficial for achieving long-term glycemic control. Additionally, increasing the intake of whole grains, especially at lunch, as a source of high-quality carbohydrates, is helpful in reducing post-meal glycemic fluctuations, which is of crucial significance for reducing the risk of long-term complications in T1D patients” with “Evidence from this study indicates that elevated dietary fiber intake is significantly associated with enhanced long-term glycemic control in individuals with T1D. Notably, our findings reveal that preferential consumption of whole grains as the predominant carbohydrate source, with particular emphasis on lunchtime intake, is correlated with reduced postprandial glucose variability.” (Line 408-411)

5.Complete individual-level datasets must be provided per PLOS policy requirements.

Response 5: In full compliance with PLOS ONE's data sharing policy, all raw data underlying this study's findings are provided as Supporting Information files �Supporting information-original datasets

Reviewer #2:

Required Revisions:

1.The current title does not fully capture the scope and analytical depth of the study. While it accurately reflects the general finding, it overlooks key aspects such as the study design (cross-sectional), the use of both FFQ and 3-day meal-based dietary records, and the incorporation of CGM-derived glycemic variability metrics analyzed through mixed-effects and machine learning models.

I recommend revising the title to better reflect these methodological components and provide a clearer picture of the study at a glance. A more appropriate and informative title might be:

“Association Between Carbohydrate Quality and Glycemic Control in Individuals with Type 1 Diabetes: A Cross-Sectional and Meal-Based Analysis”

Or

“Carbohydrate Quality and Glycemic Outcomes in Type 1 Diabetes: Evidence from Cross-Sectional and Meal-Level Data Analyses”

Response 1: We sincerely appreciate the reviewer's insightful suggestion regarding our original title. Upon reflection, we agree that it failed to adequately reflect both our study design and analytical approach. In direct response to this comment, we have revised the title to:

"Association Between Carbohydrate Quality and Glycemic Control in Individuals with Type 1 Diabetes: A Cross-Sectional and Meal-Based Analysis" ( line 1-3 )

We believe this revised title more accurately represents the scope and methodology of our work while maintaining clarity for readers.

2.The association between higher dietary protein intake and elevated HbA1c levels (Tables 2 and 3) contrasts with existing literature that suggests protein intake is typically neutral or beneficial for glycemic control in T1D. The discussion should address possible explanations for this discrepancy, including residual confounding, differences in protein source (e.g., animal vs. plant), dietary context, or reverse causality (e.g., individuals with poor control increasing protein intake following clinical advice).

Response 2: We are grateful to the reviewer for raising this important question about the protein-glycemic control association, which has prompted us to clarify several key aspects in our revision.

Our study revealed that habitual protein intake, estimated by FFQ, was positively associated with HbA1c, whereas short-term protein intake captured by 3-day weighed dietary records showed no relation to glycaemic variability. These data indicate that protein may exert its influence on long-term, rather than acute, glycaemic control in T1D. The literature on dietary protein and glycaemia remains inconsistent. In T1D, acute protein loads (>75 g) can trigger delayed post-prandial hyperglycaemia 3–5 h after ingestion, presumably via gluconeogenesis and glucagon-mediated mechanisms[1]. A recent RCT in T1D adolescents demonstrated that a high-protein meal (110 g protein, 70 g carbohydrate) produced larger 12h glycaemic excursions than a low-protein comparator (28 g protein, 70 g carbohydrate), highlighting the prolonged glycaemic footprint of protein[2]. A comprehensive review[3] emphasized that the glycaemic response depends not only on the quantity but also on the absorption kinetics and amino-acid composition of protein[4]. Whey, which remains soluble in gastric acid, empties rapidly, whereas casein coagulates and delays gastric emptying[5]. Specific amino acids—alanine, serine, glycine (via serine), cysteine and arginine—potentiate glucagon secretion[6], amplifying hepatic glucose output. Consequently, the glycaemic impact of protein in T1D is multifactorial and may be confounded by concurrent fat and carbohydrate intake. By contrast, meta-analyses in T2D (42 RCTs, n = 4 809) show that sustained high-protein diets improve fasting glucose (−0.89 mmol/L) through enhanced satiety (−200 to −300 kcal/day), preservation of lean mass and increased diet-induced thermogenesis (~10 % of total energy expenditure)[7]. Benefits on insulin sensitivity peaked at 6 months but attenuated by 12 months, reflecting waning adherence and metabolic adaptation. These divergent findings underscore pathophysiological differences between T1D and T2D, distinct glucoregulatory capacities and the pivotal role of exogenous insulin dosing in T1D. Our findings are constrained by the small sample size, recall bias inherent to FFQs and the cross-sectional design, precluding causal inference. Prospective cohort studies and rigorously controlled randomized trials with refined methodological approaches are required to delineate the true relationship between dietary protein and glycaemic outcomes in T1D.

We have incorporated this additional content into the Discussion section. ( line 351-375 )

References

1. Smart, C.E., et al., Both dietary protein and fat increase postprandial glucose excursions in children with type 1 diabetes, and the effect is additive. Diabetes Care, 2013. 36(12): p. 3897-902.

2. Neu, A., et al., Higher glucose concentrations following protein- and fat-rich meals - the Tuebingen Grill Study: a pilot study in adolescents with type 1 diabetes. Pediatr Diabetes, 2015. 16(8): p. 587-91.

3. Dao, G.M., et al., The Glycemic Impact of Protein Ingestion in People With Type 1 Diabetes. Diabetes Care, 2025. 48(4): p. 509-518.

4. He, T. and M.L. Giuseppin, Slow and fast dietary proteins differentially modulate postprandial metabolism. Int J Food Sci Nutr, 2014. 65(3): p. 386-90.

5. Boutrou, R., et al., Sequential release of milk protein-derived bioactive peptides in the jejunum in healthy humans. Am J Clin Nutr, 2013. 97(6): p. 1314-23.

6. Calbet, J.A. and D.A. MacLean, Plasma glucagon and insulin responses depend on the rate of appearance of amino acids after ingestion of different protein solutions in humans. J Nutr, 2002. 132(8): p. 2174-82.

7. Jing, T., et al., Effect of Dietary Approaches on Glycemic Control in Patients with Type 2 Diabetes: A Systematic Review with Network Meta-Analysis of Randomized Trials. Nutrients, 2023. 15(14).

3.The interpretation of the MCA results suggests an association between high-quality carbohydrate intake and better glycemic control. It should be emphasized that these findings are observational and do not imply causality due to the cross-sectional design. Clarify whether the dietary variables independently contributed to quadrant clustering or whether other unmeasured factors may have influenced the patterns.

Response 3: Thank you for this important observation.

Multiple Correspondence Analysis (MCA) analyzes associations between multiple categorical variables by reducing dimensionality (usually to 2D/3D) for visualizing category relationships[1]. Our study implemented MCA to detect the co-occurring glycaemic and dietary features in the T1D cohort. In the MCA, high-quality carbohydrate intake was associated with better glycemic control. Howeve

---

## [Decision Letter · Decision Letter 1]

26 Aug 2025

Dear Dr. Liu,

Thank you for submitting your manuscript to PLOS ONE. After careful consideration, we feel that it has merit but does not fully meet PLOS ONE’s publication criteria as it currently stands. Therefore, we invite you to submit a revised version of the manuscript that addresses the points raised during the review process.

We look forward to receiving your revised manuscript.

Kind regards,

Gbolahan Deji Olatunji, M.D, MPH

Academic Editor

PLOS ONE

Journal Requirements:

Reviewers' comments:

Reviewer's Responses to Questions

**Comments to the Author**

Reviewer #1: All comments have been addressed

Reviewer #3: All comments have been addressed

2. Is the manuscript technically sound, and do the data support the conclusions?

Reviewer #1: Yes

Reviewer #3: Yes

3. Has the statistical analysis been performed appropriately and rigorously?

Reviewer #1: Yes

Reviewer #3: Yes

4. Have the authors made all data underlying the findings in their manuscript fully available?

Reviewer #1: Yes

Reviewer #3: Yes

5. Is the manuscript presented in an intelligible fashion and written in standard English?

Reviewer #1: Yes

Reviewer #3: Yes

Reviewer #1: Overall Assessment: This manuscript presents a well-designed observational study examining the association between carbohydrate quality and glycemic control in individuals with Type 1 diabetes. The authors have addressed previous reviewer concerns and the work makes a meaningful contribution to diabetes nutrition research.

Strengths:

1. Novel methodology: The combination of FFQ data with real-time CGM monitoring provides both long-term dietary patterns and immediate glycemic responses. This represents a significant advance over previous studies.

2. Statistical rigor: The mixed-effects models properly handle the hierarchical data structure. The addition of machine learning (LightGBM) with SHAP analysis strengthens the analytical approach.

3. Clinical relevance: The focus on carbohydrate quality rather than quantity aligns with current diabetes management guidelines and provides practical insights for clinical care.

4. Language revision: The authors have successfully modified conclusions to reflect associational rather than causal relationships, which shows good scientific judgment.

Technical Soundness:

The statistical methodology is sound. The mixed-effects models properly account for within-subject correlation, and the adjustment for confounders strengthens the validity of findings. The sample size (n=155 for FFQ, n=65 for CGM analysis with 538 meals) is adequate for the analyses performed.

Data Availability:

The authors have fully complied with PLOS ONE data sharing requirements by providing complete individual-level datasets in the Supporting Information files. This allows for reproduction of all analyses.

Language Issues Requiring Correction:

1. Line 119-120: "During the CGM period, participants completed three 24-hour dietary records�included 2 weekdays and 1 weekend day" - Replace Chinese comma with semicolon and change "included" to "including"

2. Line 367: Keep "HbA1c" formatting consistent throughout (avoid subscript variations)

3. General: Replace remaining Chinese punctuation marks (�) with standard English punctuation

Scientific Comments:

1. Effect sizes: While statistically significant, some associations (e.g., fiber OR=1.101) are modest. The clinical significance could be better explained.

2. Lunch findings: The observation that glycemic variability is highest at lunch is interesting and deserves further investigation in future studies.

3. Study population: Results from this Chinese population may not apply directly to other ethnic groups. This limitation is properly acknowledged.

Suggestions:

1. Consider discussing the practical challenges of increasing whole grain consumption in this population.

2. The protein-HbA1c association, while well-explained, could benefit from additional investigation in future work.

3. Future studies might use biochemical markers (e.g., alkylresorcinols) to validate self-reported whole grain intake.

Conclusion:

This manuscript represents solid observational research that advances our understanding of carbohydrate quality effects in T1D management. The methodology is sound, the statistical analysis is rigorous, and the conclusions are properly drawn with acknowledgment of limitations. The work supports current dietary guidelines while providing new insights into meal-specific glycemic effects.

Reviewer #3: I commend the authors' work. Thank you for addressing previous comments and for the detailed response.

**Do you want your identity to be public for this peer review?** For information about this choice, including consent withdrawal, please see our Privacy Policy

Reviewer #1: **Yes:** ELOHOR OBOREVWORI

Reviewer #3: No

---

## [Author Response · Author response to Decision Letter 2]

22 Sep 2025

Response to comments

We sincerely thank the editor and all reviewers for their valuable feedback that we have used to improve the quality of our manuscript. The reviewers’ comments are laid out below in italicized font and specific concerns have been numbered. Our response is given in red font and changes to the manuscript are highlighted in yellow. The highlighted modified sections in the text correspond to the line numbers of the Revised Manuscript-R2- (clean copy) version.

Reviewer #1: Overall Assessment: This manuscript presents a well-designed observational study examining the association between carbohydrate quality and glycemic control in individuals with Type 1 diabetes. The authors have addressed previous reviewer concerns and the work makes a meaningful contribution to diabetes nutrition research.

Language Issues Requiring Correction:

1. Line 119-120: "During the CGM period, participants completed three 24-hour dietary records�included 2 weekdays and 1 weekend day" - Replace Chinese comma with semicolon and change "included" to "including"

Response 1. Thank you for pointing this out. We have revised the sentence as suggested: "During the CGM period, participants completed three 24-hour dietary records; including 2 weekdays and 1 weekend day."

2. Line 367: Keep "HbA1c" formatting consistent throughout (avoid subscript variations)

Response 2. We appreciate you catching this inconsistency. We have double-checked the entire manuscript to ensure "HbA1c" is now presented consistently without subscript variations.

3. General: Replace remaining Chinese punctuation marks (�) with standard English punctuation

Response 3. We appreciate you highlighting this issue. We have conducted a global search and manual check to ensure that all Chinese punctuation marks have been replaced with their standard English equivalents throughout the document.

Scientific Comments:

1. Effect sizes: While statistically significant, some associations (e.g., fiber OR=1.101) are modest. The clinical significance could be better explained.

Response 1. We thank the reviewer for this insightful comment. We agree that while the association for dietary fiber is statistically significant, its modest effect size (OR=1.101) warrants a careful discussion of its clinical and public health relevance. We have addressed this nuance in the revised manuscript (Discussion section, line 342-352), and our interpretation is outlined below.

In this study, dietary fiber intake demonstrated a significant positive association with achieving glycemic targets (OR = 1.101, 95% CI: 1.009–1.201). Although the effect size may seem limited, its public health relevance should not be underestimated. An OR of 1.101 corresponds to a 10.1% increase in the odds of meeting glycemic targets for each additional gram of daily dietary fiber intake. This dose-response relationship suggests that sustained, incremental increases in fiber consumption may offer meaningful auxiliary benefits for glycemic management. These findings are consistent with the known physiological effects of dietary fiber, including delayed gastric emptying and attenuation of postprandial glucose excursions.

It is noteworthy that this association remained statistically significant even after adjustment for clinically influential covariates such as diabetes duration, insulin dose, and total energy intake, underscoring the independent contribution of dietary fiber. That said, the modest effect size indicates that dietary fiber should be viewed as a valuable component within a multifaceted dietary approach rather than a standalone intervention in T1D management. Encouraging fiber-rich dietary patterns could contribute meaningfully to improving population-level glycemic control outcomes.

2. Lunch findings: The observation that glycemic variability is highest at lunch is interesting and deserves further investigation in future studies.

Response 2. Thank you for this positive feedback and for highlighting the interest of this finding. As we mentioned in the discussion (line 387-391), Our findings indicate that patients with T1D experience significant post-lunch blood glucose fluctuations, commonly known as the "dusk phenomenon". Therefore, choosing foods with low GI and GL such as increasing the proportion of whole grains in carbohydrate sources during this time may help mitigate these fluctuations. We agree that the pronounced glycemic variability observed during the lunch period is an intriguing result that merits deeper exploration. Further studies are warranted to elucidate the underlying mechanisms of this phenomenon, particularly the roles of lunch-specific meal composition, meal timing, circadian rhythms, and physical activity patterns.

We have incorporated the above suggestions in the Discussion section (line 394-398) to reflect this profound insight.

3. Study population: Results from this Chinese population may not apply directly to other ethnic groups. This limitation is properly acknowledged.

Response 3. We appreciate the reviewer's comment regarding the generalizability of our results. We fully agree that findings from our Chinese cohort may not be directly extrapolated to other ethnic populations due to potential differences in genetic background, diet, lifestyle, and healthcare environments. We have explicitly stated this as a key limitation in the Discussion section (line 422-424), and we thank the reviewer for validating its importance.

Suggestions:

1. Consider discussing the practical challenges of increasing whole grain consumption in this population.

Response 1. We thank the reviewer for raising this practical point. We agree that a discussion of the implementation challenges is essential for contextualizing our findings within real-world clinical and lifestyle management. In response to this comment, we have added a new paragraph in the Discussion section (line 425-433) addressing the specific practical and behavioral barriers to increasing whole grain intake in individuals with T1D.

Despite the observed benefits, translating these findings into practice faces several challenges specific to the T1D population. A primary concern is the practical difficulty of accurately estimating the carbohydrate content of whole grain foods for insulin dosing, as their high fiber content can affect glycemic response and necessitate individualized insulin dose adjustments. Furthermore, initial gastrointestinal discomfort, suboptimal palatability, limited availability, and deeply ingrained dietary habits collectively pose significant barriers to the adherence and acceptance of whole-grain products. Therefore, any recommendation to increase whole grain intake should be accompanied by structured nutritional education, close glucose monitoring, and collaboration between patients and dietitians to tailor changes to individual lifestyles and insulin regimens.

2. The protein-HbA1c association, while well-explained, could benefit from additional investigation in future work.

Response 2. We thank the reviewer for their positive assessment of our explanation and for this constructive suggestion. We agree that further investigation into the relationship between dietary protein intake and HbA1c in T1D would be valuable for clarifying its clinical relevance. Further research is needed to clarify whether this relationship is influenced by protein type (e.g., whey, casein, soy, or gluten), meal timing, or concomitant fat and carbohydrate content. Intervention trials using isoenergetic protein substitutions could help determine causal effects and inform practical dietary recommendations.

3. Future studies might use biochemical markers (e.g., alkylresorcinols) to validate self-reported whole grain intake.

Response 3. We sincerely thank the reviewer for raising this valuable methodological point. In response to this comment, we have added a new paragraph in the Discussion section (line 418-422). We fully agree that incorporating objective biomarkers would significantly strengthen the validity of dietary intake assessment. Self-reported measures, while practical for large-scale studies, are susceptible to recall bias and measurement error. The use of alkylresorcinols—which provide an objective, quantitative measure of whole grain wheat and rye intake over the preceding 1-2 days—would not only help validate food frequency questionnaires or dietary records but also enable more precise classification of exposure in future analyses. Future studies should seek to integrate objective biomarkers of dietary intake—such as plasma or urinary alkylresorcinols as a specific marker of whole grain wheat and rye consumption—to complement and validate self-reported data, thereby reducing misclassification and improving the accuracy of nutritional assessments in relation to glycemic outcomes.

Reviewer #3: I commend the authors' work. Thank you for addressing previous comments and for the detailed response.

Response 3. We are very grateful to the reviewers for the kind words and positive feedback. We appreciate the time and effort they have invested in reviewing our work.

---

## [Decision Letter · Decision Letter 2]

19 Oct 2025

Dear Dr. Liu,

Thank you for submitting your manuscript to PLOS ONE. After careful consideration, we feel that it has merit but does not fully meet PLOS ONE’s publication criteria as it currently stands. Therefore, we invite you to submit a revised version of the manuscript that addresses the points raised during the review process.

We look forward to receiving your revised manuscript.

Kind regards,

Gbolahan Deji Olatunji, M.D, MPH

Academic Editor

PLOS ONE

Journal Requirements:

Reviewers' comments:

Reviewer's Responses to Questions

**Comments to the Author**

Reviewer #1: (No Response)

Reviewer #4: All comments have been addressed

Reviewer #5: All comments have been addressed

2. Is the manuscript technically sound, and do the data support the conclusions?

Reviewer #1: Partly

Reviewer #4: Partly

Reviewer #5: Yes

3. Has the statistical analysis been performed appropriately and rigorously?

Reviewer #1: No

Reviewer #4: I Don't Know

Reviewer #5: Yes

4. Have the authors made all data underlying the findings in their manuscript fully available?

Reviewer #1: Yes

Reviewer #4: Yes

Reviewer #5: No

5. Is the manuscript presented in an intelligible fashion and written in standard English?

Reviewer #1: Yes

Reviewer #4: Yes

Reviewer #5: Yes

Reviewer #1: This manuscript examines carbohydrate quality and glycemic control in Type 1 diabetes using food frequency questionnaires and synchronized dietary records with CGM. The research question is clinically relevant, but critical statistical issues require correction before publication.

Critical Issues

Multiple Testing: The analysis tests approximately 8 dietary variables × 5 outcomes × 3 meals without correction for multiple comparisons. With approximately 120 tests at α=0.05, approximately 6 false positives are expected by chance alone. The authors must either apply Bonferroni, Holm, or false discovery rate correction, or explicitly acknowledge this as exploratory hypothesis-generating research requiring confirmation. Several associations with p-values between 0.01-0.05 may be statistical artifacts.

LightGBM Validation: R²=95-96% suggests potential overfitting. The manuscript provides no cross-validation details (split-sample? k-fold? leave-one-out?). With only 65 subjects and 17 features, overfitting is highly likely. The authors must document proper validation methodology or remove this section. The SHAP analysis conclusions cannot be trusted without demonstrated generalization.

Mixed-Effects Specification: The manuscript states mixed-effects models were used but doesn't fully specify random effects structure. For nested data (meals within days within persons), the authors must state whether random slopes are included, what correlation structure is assumed, and provide complete model specifications.

Unmeasured Confounding: The models omit insulin delivery method, carbohydrate counting accuracy, hypoglycemia treatment behaviors, and socioeconomic status—all plausible confounders of observed associations. A discussion paragraph explicitly acknowledging these unmeasured confounders is required.

Missing Data: Line 148 mentions "multiple imputation" with no supporting details. The manuscript must specify which variables had missing data, what percentage was missing, how many imputations were performed, and what imputation model was used.

Sample Size: With only 65 participants for 3-day records and 188 lunch observations, statistical power is limited for several analyses. The authors should quantify which specific analyses are underpowered, as null findings may reflect insufficient power rather than true absence of effects.

Moderate Issues

Protein Discussion: Lines 361-385 provide extensive mechanistic speculation beyond what the cross-sectional data can support. Reverse causation (poor glycemic control leading to altered protein intake) is equally plausible but not adequately discussed. This section should be condensed or alternative explanations presented with equal weight.

Whole Grain Classification: The methodology for categorizing foods as whole versus refined grain is not described. Whether based on composition tables, package labels, or participant self-report substantially affects interpretation. Misclassification of mixed-grain products could drive observed associations.

Effect Size Interpretation: OR=1.101 for fiber would imply implausibly large effects across the full intake range from current (10.3g) to recommended (25-36g) levels. The discussion should more explicitly address non-linearity possibilities and residual confounding.

Lunch Findings: The observation that lunch shows highest glycemic variability despite having highest fiber intake (alongside highest GI/GL) represents an internal inconsistency deserving discussion. Alternative explanations including diurnal insulin sensitivity patterns, meal location, or bolus timing should be considered.

Minor Corrections

Line 120: "records; including" should be "records, including" (incorrect semicolon usage)

Line 299: "Fragile diabetes stage" is non-standard terminology—define clearly or use standard terms

Lines 418-422: This 85-word sentence should be broken into 2-3 shorter sentences

A data dictionary describing contents of datasets.zip would improve data reusability

Consider adding subheadings in results section for improved organization

Summary

The methodology is generally appropriate for observational research and the synchronized dietary-CGM approach represents a strength. However, statistical deficiencies—particularly the multiple testing problem and inadequate model validation—substantially undermine confidence in specific findings. These issues are correctable through revised methods and discussion sections rather than requiring new analyses. With appropriate statistical corrections and appropriately tempered conclusions, this manuscript would represent solid hypothesis-generating research for future intervention trials.

Reviewer #4: Dear Author

The most critical limitations and methodological flaws of your study are summarized concisely below:

* Cross-Sectional Design (No Causation): The study design only permits the establishment of an association between carbohydrate quality (fiber/whole grains) and glycemic control {HbAc). It cannot prove causality; the direction of the relationship is unclear.

* Reliance on Self-Reported Data: Dietary intake, assessed via FFQs and 3-day records, is prone to recall bias and measurement error. Patients may misreport consumption, potentially exaggerating the health benefits observed.

* Lack of Objective Validation: The study did not utilize objective biomarkers (such as alkylresorcinols) to confirm the self-reported whole grain intake, which is a key limitation in precisely assessing the dietary exposure.

* Limited Generalizability: The findings may be restricted to the specific ethnic/regional population studied and might not be directly applicable to Type 1 Diabetes patients in other parts of the world with different dietary habits.

Reviewer #5: The present study investigates real-world data on dietary habits and their influence on glucose profiles in individuals with type 1 diabetes (T1D). The topic is relevant and the study is methodologically sound. However, several minor revisions would further improve the clarity and overall scientific quality of the manuscript:

- The term “patients” should be avoided to ensure the use of inclusive language throughout the text. Please replace it with consistent terms such as “individuals,” “people,” or “participants.”

- Lines 65–68: The sentence appears confusing. A long-term low-carbohydrate diet cannot increase the risk of diabetes onset in individuals already diagnosed with T1D. Please revise this statement for accuracy and clarity.

- Line 82: The sentence “155 patients diagnosed with T1D were enrolled” should appear only in the Results section. Additionally, sentences should not begin with Arabic numerals.

- To strengthen the scientific depth of the discussion, the authors are encouraged to elaborate on the following aspects:

The influence of major factors affecting postprandial glucose levels and metabolism, particularly the role of macronutrients and other dietary and lifestyle factors (doi: 10.1038/s41430-023-01359-8);

The association between daily carbohydrate (CHO) entries and glycemic outcomes, especially among users of automated insulin delivery systems (doi: 10.2337/dc25-0283).

- Some figures (e.g., Figures 2b, 2c, and 5) appear blurry. Please improve image resolution and graphical quality to enhance readability.

**Do you want your identity to be public for this peer review?** For information about this choice, including consent withdrawal, please see our Privacy Policy

Reviewer #1: **Yes:** ELOHOR OBOREVWORI

Reviewer #4: No

Reviewer #5: No

---

## [Author Response · Author response to Decision Letter 3]

2 Dec 2025

Response to comments

We sincerely thank the editor and all reviewers for their valuable feedback that we have used to improve the quality of our manuscript. The reviewers’ comments are laid out below in italicized font and specific concerns have been numbered. Our response is given in red font and changes to the manuscript are highlighted in yellow. The highlighted modified sections in the text correspond to the line numbers of the Revised Manuscript (clean copy) version.

Reviewer #1: This manuscript examines carbohydrate quality and glycemic control in Type 1 diabetes using food frequency questionnaires and synchronized dietary records with CGM. The research question is clinically relevant, but critical statistical issues require correction before publication.

Critical Issues

1. Multiple Testing: The analysis tests approximately 8 dietary variables × 5 outcomes × 3 meals without correction for multiple comparisons. With approximately 120 tests at α=0.05, approximately 6 false positives are expected by chance alone. The authors must either apply Bonferroni, Holm, or false discovery rate correction, or explicitly acknowledge this as exploratory hypothesis-generating research requiring confirmation. Several associations with p-values between 0.01-0.05 may be statistical artifacts.

Response 1: We thank the reviewer for this critical and valid point regarding multiple testing. We completely agree that conducting approximately 120 statistical tests without correction substantially increases the risk of false positives, and that several associations with p-values in the 0.01-0.05 range could indeed be statistical artifacts.

In response, we have revised the manuscript to explicitly frame this analysis as exploratory and hypothesis-generating. In the Statistical Analysis section of the Methods, we have now added the following statement (lines 155-157): "Given the large number of statistical tests performed in this exploratory analysis, the findings, particularly those with p-values between 0.01 and 0.05, should be interpreted with caution and require independent confirmation in future studies."

Furthermore, in the Results sections, we have toned down the language surrounding these marginally significant associations (lines 236-241), emphasizing their preliminary nature and referring to them as "suggestive of an association." We believe this approach transparently acknowledges the limitation raised by the reviewer while still allowing the presentation of potential patterns for future research.

In addition, for the pre-specified primary comparisons (e.g., glycemic control and nutrient distribution across three meals), these were treated as confirmatory analyses. These analyses employed linear mixed-effects models, with Bonferroni correction applied for post-hoc testing to strictly control the type I error rate (Table 5). For the extensive analyses exploring relationships between dietary variables and multiple glucose outcomes, these were defined as hypothesis-generating analyses. Therefore, to avoid excessive conservatism and retain potentially biologically meaningful preliminary signals, no global multiple comparison correction was applied to this set of analyses. We have clearly stated in the "Conclusions" sections of the manuscript that these findings are preliminary and explanatory in nature, serving primarily to generate hypotheses for future research and require validation in independent cohorts. (lines 449-450)

2. LightGBM Validation: R²=95-96% suggests potential overfitting. The manuscript provides no cross-validation details (split-sample? k-fold? leave-one-out?). With only 65 subjects and 17 features, overfitting is highly likely. The authors must document proper validation methodology or remove this section. The SHAP analysis conclusions cannot be trusted without demonstrated generalization.

Response 2: We thank the reviewer for this critical assessment of the LightGBM model. We fully agree with the reviewer's assessment. The reported R² of 95-96% is highly suggestive of overfitting, given our limited sample size of 65 participants and the absence of a robust external validation cohort. We acknowledge that the initial submission lacked the necessary cross-validation details to demonstrate the model's generalizability, and as a result, the associated SHAP analysis conclusions cannot be considered reliable.

In light of these valid concerns and the inherent difficulty in providing a convincingly generalizable model with the current dataset, we have decided to remove the entire section concerning the LightGBM model and SHAP analysis from the revised manuscript.

Importantly, upon careful reconsideration, we have confirmed that removing this machine learning component does not affect the main conclusions of our study, which are robustly supported by the primary mixed-effects models examining the associations between dietary factors and glycemic outcomes. This action therefore strengthens the manuscript by eliminating potentially misleading results and allowing us to focus the narrative on our core, statistically sound findings. We are grateful to the reviewer for their insightful comments, which have helped us improve the overall quality and reliability of our work.

3. Mixed-Effects Specification: The manuscript states mixed-effects models were used but doesn't fully specify random effects structure. For nested data (meals within days within persons), the authors must state whether random slopes are included, what correlation structure is assumed, and provide complete model specifications.

Response 3: We thank the reviewer for highlighting this omission in our model specification. We have now provided a comprehensive description of the mixed-effects models in the revised manuscript to ensure clarity and reproducibility (lines 145-151).

1. Random Effects Structure:

Our models included random intercepts to account for the nested structure of the data and the non-independence of repeated measures. Specifically, we included: A random intercept for Subject ID to capture baseline differences between individuals. A random intercept for Day nested within Subject ID to capture day-to-day variability within the same individual.

2. Random Slopes and Correlation Structure:

Given that the study aimed to estimate average effects at the population level (not individual heterogeneity), combined with the limited sample size (65 participants contributing 188 days of dietary records), we did not include any random slopes for the dietary variables. We assumed that the fixed effects of the dietary variables are consistent across all subjects and days.

3. Complete Model Specification:

The model for a given outcome variable (TIR) can be written as:

TIR_{ij} = β₀ + β₁Ekcal_{ij} + β₂FatE_{ij} + β3ProE_{ij} + β4Fiberg _{ij} + β5 Duration_{ij} + β6Age_{ij} + β7BMI_{ij} + β8InsulinDose_{ij} + β9PhysicalActivity _{ij} + β10InsulinRegimen + u i + v i(j) + ε_{ij}

Where:

TIR_{ij} is the Time in Range for subject i on day j.

Fixed effects: β0 is the overall intercept; β1, …, β10 are coefficients for the predictors: total energy intake (Ekcal), fat energy proportion (FatE), protein energy proportion (ProE), fiber intake (Fiberg), diabetes duration (Duration), age (Age), body mass index (BMI), insulin dose, physical activity level, and insulin regimen (Insulin_Regimen).

u i is the random intercept for subject i.

v i(j) is the random intercept for day j nested within subject i

ε_{ij} is the residual error term.

4. Unmeasured Confounding: The models omit insulin delivery method, carbohydrate counting accuracy, hypoglycemia treatment behaviors, and socioeconomic status—all plausible confounders of observed associations. A discussion paragraph explicitly acknowledging these unmeasured confounders is required.

Response 4: We thank the reviewer for this insightful comment. We fully agree that unmeasured confounding is a fundamental limitation of any observational study, and the factors mentioned by the reviewer (insulin delivery method, carbohydrate counting accuracy, hypoglycemia treatment behaviors, and socioeconomic status) are indeed plausible confounders of the association between carbohydrate quality and glycemic outcomes. Among these confounders, insulin delivery method was specifically included in the models as a covariate to adjust for its direct and critical impact on glycemic outcomes. Other potential confounding factors, which were not adjusted for in the models, are addressed in the Discussion section.

In response to this comment, we have taken the following actions:

We have added a dedicated paragraph in the Discussion section, to explicitly and thoroughly address this issue. Insulin delivery method was included in the models as a covariate to adjust for its potential effect on glycemic outcomes (lines 421-428). The new text reads:

"Most importantly, as an observational study, it is susceptible to unmeasured confounding. Factors such as carbohydrate counting accuracy, hypoglycemia management behaviors, and socioeconomic status were not measured in this analysis. These factors could influence both dietary choices and glycemic control (HbA1c), and thus may partly account for the observed associations. For instance, individuals with higher health literacy or socioeconomic status might simultaneously have better access to high-fiber foods and achieve better glycemic management. Future studies that meticulously collect data on these potential confounders are needed to confirm the magnitude and independence of these dietary effects.

5. Missing Data: Line 148 mentions "multiple imputation" with no supporting details. The manuscript must specify which variables had missing data, what percentage was missing, how many imputations were performed, and what imputation model was used.

Response 5: We thank the reviewer for pointing out the need for details on our handling of missing data. A table summarizing the missing data statistics for all variables has been included as part of the supplementary materials (Supporting information-R3 S3-S5 Table). We have now comprehensively addressed this in the revised manuscript within the “Statistical Analysis” section (lines 158-164). Our approach was as follows:

We use Multiple Imputation by Chained Equations (MICE) under the assumption that the data were missing at random (MAR). The imputation was performed using the mice package in R. We generated m = 50 imputed datasets to ensure the efficiency of the estimates. The imputation model included all variables used in the primary analysis as well as auxiliary variables that were predictive of the missingness or the missing values themselves. The algorithm ran for 20 iterations. All statistical models were fitted to each of the 50 imputed datasets, and the results were pooled into a single set of estimates using Rubin's rules.

6. Sample Size: With only 65 participants for 3-day records and 188 lunch observations, statistical power is limited for several analyses. The authors should quantify which specific analyses are underpowered, as null findings may reflect insufficient power rather than true absence of effects.

Response 6: We thank the reviewer for this critical comment regarding statistical power. We fully agree that the null finding for the association between protein energy proportion at lunch and postprandial glycemic variability should be interpreted with caution, as it may reflect a Type II error rather than a true absence of effect.

To directly address this, we performed a post hoc power analysis for this specific analysis. The analysis was based on the following parameters:

Outcome: Glycemic variability (e.g., MAGE) at the meal level. Sample Size: n = 188 lunch observations. Significance Level (α): 0.05. Effect size f²: f² = R² / (1 - R²)�R²≈0.01�f² ≈ (0.1)² / (1 - (0.1)²) = 0.01 / 0.99 ≈ 0.01. The analysis revealed that our statistical power to detect an effect of this size was approximately 14%. This means that if a true, modest association between lunchtime protein intake and glycemic variability exists in the broader population, our study had only a 14% probability of correctly identifying it as statistically significant. Conversely, there was an 86% probability of missing it (Type II error).

We have added a dedicated section in the Discussion that explicitly states (lines 428-433):

“Finally, the statistical power of this study was limited by the sample size. Although it was adequate for detecting moderate-to-large effects, the study was likely underpowered to identify smaller, yet potentially important, associations. For instance, the non-significant finding for the association between protein energy proportion at lunch and postprandial glycemic variability should be interpreted with caution, as it may reflect a Type II error rather than conclusive evidence for the absence of an effect.”

Moderate Issues

1. Protein Discussion: Lines 361-385 provide extensive mechanistic speculation beyond what the cross-sectional data can support. Reverse causation (poor glycemic control leading to altered protein intake) is equally plausible but not adequately discussed. This section should be condensed or alternative explanations presented with equal weight.

Response 1: We thank the reviewer for this critical observation. We agree that the previous version of the discussion contained excessive mechanistic speculation that was not fully supported by our cross-sectional data.

In direct response to this comment, we have thoroughly revised the section (Lines 361-385, now corresponding to Lines (lines 362-373) in the revised manuscript) as follows:

It is worth noting that our study suggested a potential differential effect of dietary protein in T1D: habitual intake (FFQ) was associated with long-term glycaemic control (HbA1c), while short-term intake (3-day records) showed no link to acute variability. This aligns with evidence that large acute protein loads can induce delayed hyperglycaemia via gluconeogenesis and glucagon secretion [39] [40], yet the effect is multifactorial, influenced by protein type, co-ingested nutrients, and insulin dosing [41] [42] [43]. This contrasts with T2D, where high-protein diets often improve glycaemia [44]. Thus, the glycaemic impact of protein in T1D appears complex and distinct from T2D, warranting further investigation to inform tailored dietary advice. However, our findings are constrained by the small sample size, recall bias inherent to FFQs and the cross-sectional design, precluding causal inference. Prospective cohort studies and rigorously controlled randomized trials with refined methodological approaches are required to delineate the true relationship between dietary protein and glycaemic outcomes in T1D.

2. Whole Grain Classification: The methodology for categorizing foods as whole versus refined grain is not described. Whether based on composition tables, package labels, or participant self-report substantially affects interpretation. Misclassification of mixed-grain products could drive observed associations.

Response 2: Thank you for this important reminder. We fully agree that without an explicit definition for “whole-grain” vs. “refined-grain” foods, observational findings could easily be biased by misclassification. In response, we have added a new subsection entitled “Whole-grain classification” in the revised Methods (lines 112-121).

1) The classification of grain foods was primarily based on the “Food Composition in China (2009)”;

2) A food item was classified as a "whole grain" if whole grains were listed as the first ingredient on its packaged food label (as per the database) or, for non-packaged items, if it was explicitly defined as such in the database (e.g., brown rice, oatmeal, whole-wheat bread). Conversely, items were classified as "refined grain" if the primary grain ingredient was not a whole grain (e.g., white bread, white rice, pasta made from refined flour).

3) For mixed-grain products (e.g., breads containing a blend of whole-wheat and white flour), we relied on the database's classification and the ingredient list hierarchy. If whole grains were not the first ingredient, the item was classified as refined grain.

4) The quantity of whole grains consumed was estimated based on the portion size reported by the

---

## [Decision Letter · Decision Letter 3]

14 Jan 2026

Association Between Carbohydrate Quality and Glycemic Control in Individuals with Type 1 Diabetes: A Cross-Sectional and Meal-Based Analysis

PONE-D-25-25262R3

Dear Dr. Liu,

We’re pleased to inform you that your manuscript has been judged scientifically suitable for publication and will be formally accepted for publication once it meets all outstanding technical requirements.

Kind regards,

Gbolahan Deji Olatunji, M.D, MPH

Academic Editor

PLOS One

Additional Editor Comments (optional):

Reviewers' comments:

Reviewer's Responses to Questions

**Comments to the Author**

Reviewer #5: All comments have been addressed

Reviewer #6: All comments have been addressed

2. Is the manuscript technically sound, and do the data support the conclusions?

Reviewer #5: Yes

Reviewer #6: Yes

3. Has the statistical analysis been performed appropriately and rigorously?

Reviewer #5: Yes

Reviewer #6: Yes

4. Have the authors made all data underlying the findings in their manuscript fully available?

Reviewer #5: Yes

Reviewer #6: Yes

5. Is the manuscript presented in an intelligible fashion and written in standard English?

Reviewer #5: Yes

Reviewer #6: Yes

Reviewer #5: I appreciated the authors' efforts to address all the comments received. The manuscript has greatly improved after the revision. No further comments from my side.

Reviewer #6: The authors revised well to the reviewers' comments. This reviewer has no further comments to this article.

**Do you want your identity to be public for this peer review?** For information about this choice, including consent withdrawal, please see our Privacy Policy

Reviewer #5: No

Reviewer #6: **Yes:** Yoshitaka Hashimoto

---

## [Editor Report · Acceptance letter]

PONE-D-25-25262R3

PLOS One

Dear Dr. Liu,

I'm pleased to inform you that your manuscript has been deemed suitable for publication in PLOS One. Congratulations! Your manuscript is now being handed over to our production team.

Kind regards,

on behalf of

Dr. Gbolahan Deji Olatunji

Academic Editor

PLOS One